# Experimental vaccination by single dose sporozoite injection of blood-stage attenuated malaria parasites

Julia M Sattler [1,2], Lukas Keiber[1,2,6], Aiman Abdelrahim [1,2,6], Xinyu Zheng[1,2], Martin Jäcklin[1,2], Luisa Zechel[1,2], Catherine A Moreau[1], Smilla Steinbrück[1,2], Manuel Fischer[3], Chris J Janse [4], Angelika Hoffmann[3,5], Franziska Hentzschel [1,2] & Friedrich Frischknecht [1,2]✉

## Abstract

Malaria vaccination approaches using live *Plasmodium* parasites are currently explored, with either attenuated mosquito-derived sporozoites or attenuated blood-stage parasites. Both approaches would profit from the availability of attenuated and avirulent parasites with a reduced blood-stage multiplication rate. Here we screened gene-deletion mutants of the rodent parasite *P. berghei* and the human parasite *P. falciparum* for slow growth. Furthermore, we tested the *P. berghei* mutants for avirulence and resolving blood-stage infections, while preserving sporozoite formation and liver infection. Targeting 51 genes yielded 18 *P. berghei* gene-deletion mutants with several mutants causing mild infections. Infections with the two most attenuated mutants either by blood stages or by sporozoites were cleared by the immune response. Immunization of mice led to protection from disease after challenge with wild-type sporozoites. Two of six generated *P. falciparum* gene-deletion mutants showed a slow growth rate. Slow-growing, avirulent *P. falciparum* mutants will constitute valuable tools to inform on the induction of immune responses and will aid in developing new as well as safeguarding existing attenuated parasite vaccines.

**Keywords** Genetic Attenuation; Malaria; *Plasmodium*; Vaccine; Virulence
**Subject Categories** Immunology; Microbiology, Virology & Host Pathogen Interaction

## Introduction

Malaria is caused by the cyclical infection of red blood cells by *Plasmodium* parasites, which undergo a complex life cycle (Fig. 1A) and are first injected into the vertebrate by a mosquito bite. During the bite, *Plasmodium* sporozoites are deposited in the skin where they rapidly migrate to enter blood vessels. After being transported to the mammalian liver, they enter hepatocytes and develop into merozoites, which invade red blood cells (RBC). Different parasite life cycle stages, such as sporozoites, liver- and blood stages, can induce immune responses, which have been exploited for the generation of subunit or whole parasite vaccines (Good and Stanisic, 2020; Kurtovic et al, 2021; Nevagi et al, 2021). The most advanced subunit malaria vaccine developed to date, RTS,S (Mosquirix) offers 30–50% protection against clinical malaria (RTS,S Clinical Trials Partnership et al, 2012; RTS,S Clinical Trials Partnership, 2015) with the newer R21 recently showing around 75% protection (Datoo et al, 2021). These subunit vaccines are composed of part of the *Plasmodium* circumsporozoite protein, the most abundant protein on the surface of the sporozoite (Swearingen et al, 2016; Lindner et al, 2019). Several other subunit vaccines are also being developed and evaluated (Tiono et al, 2018; Mordmüller et al, 2019; Minassian et al, 2021; Rosenkranz et al, 2023).

The use of attenuated parasites as vaccines has also been explored in the forms of attenuated blood-stages (whole blood-stage; Wbs) and sporozoites attenuated in liver-stage development (whole sporozoite; Wsp) (Fig. 1B). Attenuation of parasites to generate whole-organism vaccines can be achieved by irradiation, application of chemicals or genetic modification (Nussenzweig et al, 1967; Mueller et al, 2005; Purcell et al, 2008; Mordmüller et al, 2017; Roestenberg et al, 2020; Murphy et al, 2022; Goswami et al, 2024). The most advanced Wsp vaccine candidate, the PfSPZ Vaccine, consisting of cryopreserved, vialed radiation-attenuated sporozoites has entered phase 3 clinical development (Seder et al, 2013; Ishizuka et al, 2016; Epstein et al, 2017; Mordmüller et al, 2017; Lyke et al, 2017; Sissoko et al, 2022; Sirima et al, 2022). These sporozoites enter hepatocytes but are unable to replicate and thus abort development early in the liver. Sporozoite attenuation by genetic modification rather than by radiation offers the advantage of a more homogeneous product, increased biosafety for sporozoite production and for some genetically attenuated parasites (GAP) increased potency (Butler et al, 2011). Several genetically attenuated Wsp vaccines have been developed by the deletion of one to three genes and have been evaluated for inducing protective immune responses in clinical trials (Kublin et al, 2017; Roestenberg et al,

[1]Integrative Parasitology, Center for Infectious Diseases, Heidelberg University Medical School, 69120 Heidelberg, Germany. [2]German Center for Infection Research, DZIF, Partner Site Heidelberg, Heidelberg, Germany. [3]Department of Neuroradiology, Heidelberg University Medical School, 69120 Heidelberg, Germany. [4]Leiden Malaria Research Group, Parasitology, Leiden University Medical Center, Leiden, The Netherlands. [5]Present address: Department of Neuroradiology, University Institute of Diagnostic and Interventional Neuroradiology, University Hospital Bern, Inselspital, University of Bern, 3010 Bern, Switzerland. [6]These authors contributed equally: Lukas Keiber, Aiman Abdelrahim. ✉E-mail: freddy.frischknecht@med.uni-heidelberg.de

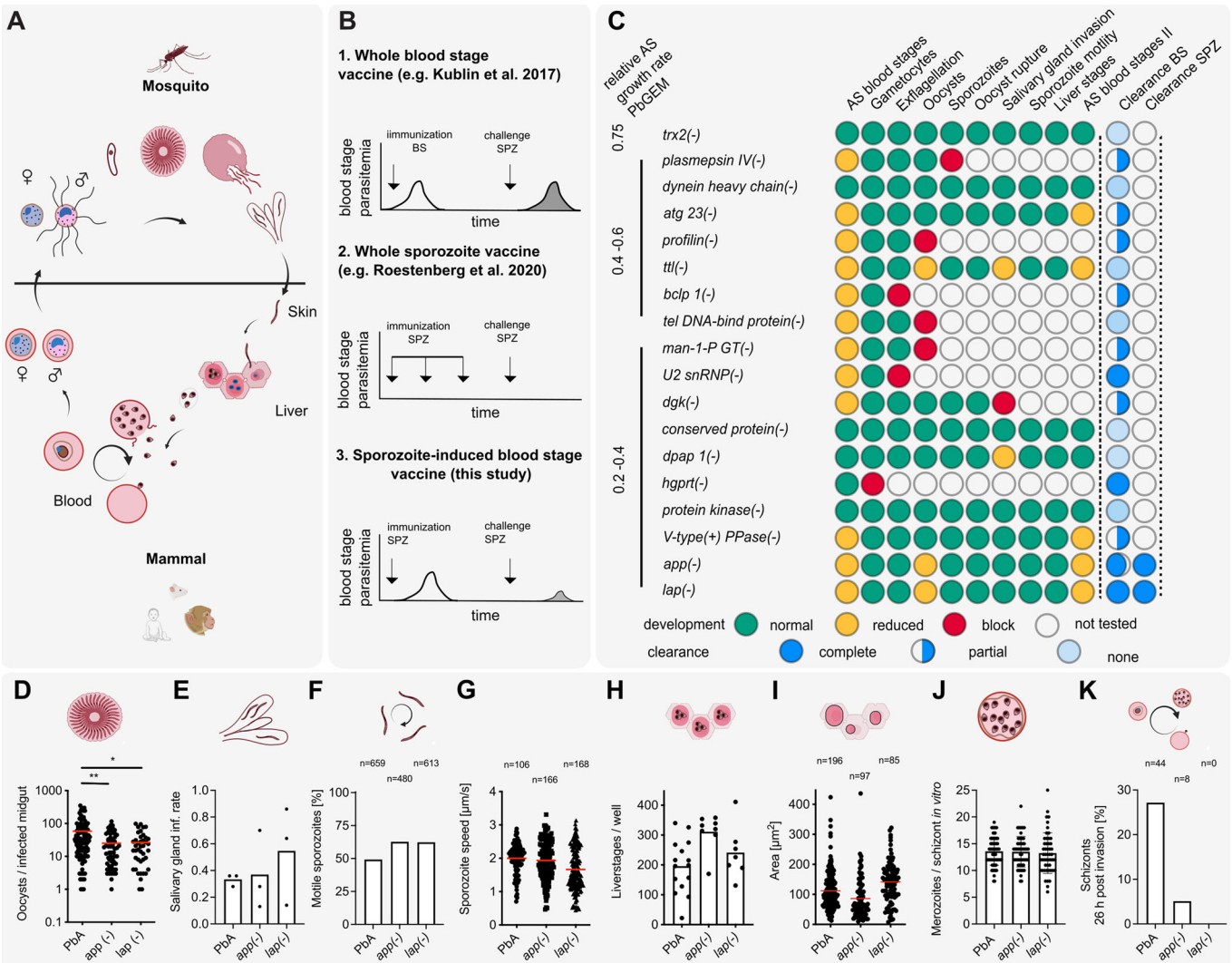

**Figure 1. Screening for *P. berghei* gene-deletion mutants with slow-growing blood-stages that retain mosquito transmissibility.**

(A) Simplified *Plasmodium* life cycle indicating the stages investigated in this report. (B) Schematics of genetic attenuation strategies. Mutant parasites can be generated to (i) attenuate in the blood-stage, (ii) enter into the liver to arrest their development and, as shown here (iii) complete the life cycle but attenuate in the blood-stage. Immunization schemes and model blood-stage parasitemia curves upon immunization and challenge are shown. (C) Life cycle progression analysis of gene-deletion mutant lines lacking the indicated genes. Top to bottom: fastest to slowest blood-stage growth rate of the parasite lines. Green and yellow circles: comparable and reduced developmental characteristics, each compared to *Plasmodium berghei* strain Anka (PbA) wild-type. Red circles: a complete block in life cycle progression. Blue circles: relative level of clearance of blood-stages after infection by 100 iRBC i.v. or by sporozoite-induced infections (1000 spz *i.v.* or natural transmission) in SWISS and C57BL/6 mice. AS: asexual, BS: blood-stage, SPZ: sporozoites. (D) Oocyst numbers in mosquitoes infected with wild-type (*Pb*A), *app*(−), and *lap*(−). * and ** indicate *P* < 0.05 and *P* < 0.01, respectively; a Shapiro–Wilk normality test was applied followed by a Kruskall–Wallis test. Three independent infections were analyzed. (E) Salivary gland invasion in mosquitoes infected with wild-type (*Pb*A), *app*(−), and *lap*(−) mutants. The ratio of salivary gland- versus midgut-derived sporozoites are shown from three independent experiments. (F, G) Wild-type-like range of sporozoite gliding pattern (F) and motility speed (G) of the gene-deletion parasite lines; n: number of sporozoites analyzed. (H, I) Wild-type-like range of in vitro liver cell infection (H) and liver-stage development (I) as evaluated 48 h post-infection of HepG2 cells. (H) Three independent experiments with multiple replicates; (I) *n* indicates the number of liver-stages analyzed. (J) Number of merozoites per schizont in in vitro cultured, mature schizonts of wild-type (*Pb*A), *app*(−), and *lap*(−) parasites. n: number of schizonts analyzed. (K) Percentage of iRBCs that developed to schizonts at 26 h post merozoite invasion in wild-type (*Pb*A) or *app*(−) and *lap*(−) mutant parasite lines, indicating delayed blood-stage development of *app*(−) and *lap*(−) parasites. *n* indicates the number of schizonts detected in three independent experiments. Source data are available online for this figure.

2020; Murphy et al, 2022). The studies using Wsp immunization revealed that Wsp can induce levels of protection that are higher than those achieved with various subunit malaria vaccines currently under investigation (Good and Stanisic, 2020).

In addition to attenuated Wsp, vaccination approaches using Wbs have also been explored (Stanisic et al, 2018; Good and

Stanisic, 2020; Stanisic et al, 2021; Webster et al, 2021). However, progress towards a Wbs vaccine is much less advanced compared to Wsp vaccines. Immunization with Wbs has been performed with chemically attenuated (Stanisic et al, 2021; Nevagi et al, 2021; Good and Stanisic, 2020) and genetically attenuated Wbs (Ting et al, 2008; Spaccapelo et al, 2010; Aly et al, 2010) in preclinical studies

using rodent malaria models. Immunization with genetically attenuated Wbs of rodent malaria parasites with a reduced growth rate is known to induce robust protection against wild-type challenge in mice (Ting et al, 2008; Spaccapelo et al, 2010; Aly et al, 2010; Mathieu et al, 2015; Demarta-Gatsi et al, 2016), but in contrast to the clinically silent Wsp, Wbs immunization (without also applying drug treatment) harbors the risk of causing a symptomatic blood-stage infection. Therefore, a high degree of attenuation (and absence of virulence) during blood-stage growth is essential. In one study, immunization with genetically modified Wbs of *P. falciparum* has been evaluated in humans. In these Wbs a gene has been deleted that encodes the knob-associated histidine-rich protein (KAHRP), which is responsible for the assembly of knob structures at the infected erythrocyte membrane (Webster et al, 2021). Knobs are required for correct display of the polymorphic adhesion ligand *P. falciparum* erythrocyte membrane protein 1 (PfEMP1), a key virulence determinant (Crabb et al, 1997). Although the Wbs were immunogenic, administration of higher Wbs doses resulted in blood-stage infections with significant parasitemias and malaria-associated symptoms (Webster et al, 2021).

For further development of safe Wbs vaccines, it is important to explore the generation of genetically attenuated Wbs that are avirulent, for example, Wbs with a strongly reduced multiplication rate that result in mild infections that resolve even in the absence of drug treatment. Such Wbs could be used to study the immune response to controlled human infections and possibly employed for repeated vaccinations to build up blood-stage immunity in analogy to naturally occurring immunity in highly endemic regions (Doolan et al, 2009).

In this study, we have generated and screened gene-deletion mutant parasites in the rodent parasite *P. berghei* and the human parasite *P. falciparum* for slow blood-stage multiplication rates. In addition, we screened the *P. berghei* mutants for avirulence in mice, i.e., absence of experimental cerebral malaria (ECM), capacity to be transmitted by mosquitoes and for resolving blood-stage infections. Two of the eighteen screened *P. berghei* mutants showed these characteristics and conferred protection from wild-type challenge after single low dose sporozoite immunization. Two of six generated *P. falciparum* gene-deletion mutants showed a slow growth rate. Slow-growing, avirulent *P. falciparum* mutants will constitute valuable tools to test induction and breadth of human immune responses and may aid in developing genetically attenuated vaccines to build up liver- and blood-stage immunity as well as to limit the danger of unwanted virulent blood-stage infections of Wsp vaccines.

## Results

### Screening for *P. berghei* gene-deletion mutants with slow blood-stage growth for mosquito transmissibility

For the creation of mutants with slow-growing blood stages, we took advantage of the data available from the *Plasmodium* genetic modification project (PlasmoGEM) (Gomes et al, 2015). In a large-scale genetic screen (Bushell et al, 2017) blood-stage growth rates were determined for over 2000 nonclonal gene-deletion mutants. From this screen, we selected 47 gene deletions with a predicted

reduced growth rate either by 40–60% (set 1, 11 genes) or by 60–80% (set 2, 36 genes). In addition, we selected one gene-deletion mutant, *thioredoxin 2* (*trx2*), with a predicted growth rate that was reduced by 25%. For the 48 selected genes, we used the gene-deletion DNA constructs available from the PlasmoGem resource to generate and select clonal mutant parasites (Fig. 1C; Appendix Figs. S1 and S2 and Appendix Tables S1–S4). Of the 48 targeted genes, we were able to select populations of mixed wild-type and transgenic parasites for 24 genes (Appendix Table S1). From those populations, we could isolate transgenic clonal lines for 15 genes. Of these genes, one was *trx2*, 6 genes were from set 1 and 8 genes from set 2 (Fig. 1C; Appendix Table S2). We further tested three slow-growing parasite lines lacking *plasmepsin IV* (*pmIV*), *aminopeptidase p* (*app*) and *M17 leucyl-aminopeptidase* (*lap*) (Spaccapelo et al, 2010; Lin et al, 2015). Around half of the in total 18 clonal mutant parasite lines showed reduced blood-stage growth rates comparable to those reported in the PlasmoGEM screen (Bushell et al, 2017) with narrow confidence intervals of PlasmoGEM growth rates being a predictor for the growth rate of clonal lines (Appendix Table S5 and Appendix Fig. S3). The other half showed growth rates closer to the growth rate of wild-type parasites.

We next analyzed life cycle progression of the 18 gene-deletion mutants through mosquitoes. Seven mutants showed defects before sporozoite formation. These mutants lacked the genes encoding plasmepsin IV (PMIV), profilin, beta-catenin like protein 1 (BCLP1), telomeric DNA-binding protein (tel DNA-bind protein), mannose-1-phosphate guanyltransferase (Man-1-P GT), U2 snRNP-associated SURP motif-containing protein (U2 snRNP) and hypoxanthine-guanine phosphoribosyl transferase (HGPRT) (Fig. 1C; Appendix Table S5). Deletion of the gene *diacylglycerol kinase (DGK)* produced mutant parasites that successfully infected mosquitoes and produced midgut sporozoites but these were unable to invade salivary glands. Notably, the other ten mutant lines were able to complete the life cycle (Fig. 1C–I; Appendix Fig. S4). These mutants lacked the genes encoding TRX2, dynein heavy chain, autophagy-related protein 23 (ATG 23), tubulin tyrosine ligase (TTL), a "conserved protein", dipeptidyl amino-peptidase 1 (DPAP 1), protein kinase, V-type(+) pyrophosphatase (V-type(+) PPase), APP or LAP. The latter two proteins are proteases involved in hemoglobin digestion (Dalal and Klemba, 2007; Lin et al, 2015).

In mosquitoes, even the two slowest replicating gene-deletion mutants, *app(−)* and *lap(−)*, developed oocysts, albeit at somewhat lower levels than wild-type (Fig. 1D), resulting in correspondingly lower sporozoite numbers in salivary glands (Fig. 1E; Appendix Table S5). Salivary gland sporozoites of *app(−)* and *lap(−)* mutants showed motility comparable to wild-type sporozoites (Fig. 1F,G) and infected and developed in in vitro cultured hepatocytes similar to wild-type parasites (Fig. 1H,I). To assess if the reduced blood-stage growth of *app(−)* and *lap(−)* parasites (Lin et al, 2015) was due to a reduced number of merozoites produced per intraery-throcytic cycle, we cultivated iRBCs overnight in vitro, which allows parasites to develop into mature schizonts. Quantifying the number of merozoites per schizont showed no difference between the wild-type and the mutants (Fig. 1J). However, this result does not rule out a potential slower intraerythrocytic development. To assess this, we injected mature schizonts into naive mice to synchronize infection and harvested blood again once ring-stage

**Table 1. Summary of PbA wild type, app(-) and lap(-) infections of SWISS and C57BL/6 mice.**

| | 100 iRBC SWISS | | |
|---|---|---|---|
| **A** | **Mice clearing/ mice infected (prepatency [d])** | **Peak parasitemia [%] (d)** | **Blood stage cleared (d)** |
| PbA | 0/3 (6) | 27 | n.a. |
| *app(-)* | 3/4 (5) | 22 (18) | 21 |
| *lap(-)* | 4/4 (6) | 17 (18) | 21 |
| | 100 iRBC C57BL/6 | | |
| | **Mice clearing/ mice infected (prepatency [d])** | **Peak parasitemia [%] (d)** | **Blood stage cleared (d)** |
| PbA | 0/4 (4) | 4 | n.a. |
| *app(-)* | 4/4 (6) | 9 (14) | 21 |
| *lap(-)* | 4/4 (5) | 11 (15) | 20 |
| | Natural transmission C57BL/6 | | |
| **B** | **Mice clearing/ mice infected (prepatency [d])** | **Peak parasitemia [%] (d)** | **Blood stage cleared (d)** |
| PbA | 0/4 (3) | 6 | n.a. |
| *app(-)* | 7/8 (5) | 12 (14) | 22 |
| *lap(-)* | 8/8 (6) | 16 (15) | 22 |
| | 1000 salivary gland sporozoites C57BL/6 i.v. | | |
| | **Mice clearing/ mice infected (prepatency [d])** | **Peak parasitemia [%] (d)** | **Blood stage cleared (d)** |
| PbA | 0/8 (4) | 4 | n.a. |
| *app(-)* | 24/24 (6) | 19 (17) | 27 |
| *lap(-)* | 24/24 (6) | 19 (17) | 23 |
| | 10,000 salivary gland sporozoite SWISS i.v. | | |
| | **Mice clearing/ mice infected(prepatency [d])** | **Peak parasitemia [%] (d)** | **Blood stage cleared (d)** |
| PbA | 0/6 (5) | 23 | n.a. |
| *app(-)* | 6/6 (7) | 14 (16) | 23 |
| *lap(-)* | 4/4 (7) | 7 (16) | 20 |

*n.a.* not applicable.
(A) Blood-stage-initiated infections by intravenous injection of 100 infected red blood cells (iRBC) (Fig. 2A–C); (B) mosquito stage-initiated infections by the bite of 10 mosquitoes per mouse (natural transmission) or the intravenous injection of 1000 or 10,000 sporozoites (Fig. 2D–I). Numbers for peak parasitemia and days are averages from the indicated number of investigated mice.

parasites developed to assay the maturation of the ensuing blood-stage in vitro. Culturing these ring-stages for 26 h revealed a reduced proportion of mature schizonts in *app(−)* and *lap(−)* parasites compared to wild-type, with *lap(−)* parasites being morphologically severely altered and appearing immature by Giemsa staining (Fig. 1K). Overall, these results suggest that *app(−)* and *lap(−)* are attenuated due to a prolonged intraery-throcytic developmental time.

## Blood-stage-induced infections of five gene-deletion mutants are resolved by mice

To determine virulence (i.e., death or induction of experimental cerebral malaria; ECM) of the 18 gene-deletion mutants, we infected outbred SWISS mice or inbred, highly ECM-susceptible

C57BL/6 mice by intravenous injection of 100 blood-stages. Infection of SWISS mice led to a delayed prepatency for 5 of the 18 mutants lacking genes encoding Trx2, Profilin, U2 snRNP, HGPRT, and V-type(+) PPase. Growth of these five mutants led to the death of the mice or was controlled, and parasites were cleared in all or in a subset of mice (Fig. 1C; Table EV1, Appendix Figs. S5 and S6, and Appendix Table S6). The three mutants with complete clearance lacked the genes encoding LAP, U2 snRNP, and HGPRT (Table EV1; Appendix Figs. S5 and S6 and Appendix Table S6). In SWISS mice infected with *app(−)* or *lap(−)* blood stages, parasitemia peaked on average on day 18 and was resolved by day 21 (Table 1; Table EV1 and Appendix Figs. S5 and S6). Out of 4 *app(−)* blood-stage infected mice only one SWISS mouse died due to anemia at day 21 post infection (Table 1; Appendix Fig. S6 and Appendix Table S6). In contrast, all 4 SWISS mice infected with *lap(−)* blood-stages were able to clear the infection (Table 1; Appendix Fig. S6 and Appendix Table S6). Infections with wild-type or the other gene-deletion mutants were not completely resolved and resulted in death of some or all animals (Fig. 1C; Table 1, Appendix Figs. S5 and S6, and Appendix Table S6). In C57BL/6 mice, infection with 100 iRBC led for all nine tested mutants to a delayed infection that could be fully controlled and cleared by the mice infected with *U2 snRNP(−)*, *dgk(−)*, *hgprt(−)*, *app(−)*, and *lap(−)* parasites (Figs. 1C and 2A–C; Table EV1, Appendix Fig. S7 and Appendix Table S6). As illustrated for *app(−)* or *lap(−)* blood-stage-induced infections of C57BL/6 mice, the infection peaked around day 14 and 15, respectively, and was ultimately resolved within 20–21 days (Fig. 2A–C; Table 1; Table EV1) based on Giemsa-stained blood smears. To investigate sub-patent persistent infections we performed qPCR on a subset of these *lap(−)*-infected mice. This showed a prolonged infection compared to Giemsa-based detection but also complete clearance of the parasites from blood (Appendix Fig. S8). However, we cannot rule out a potential hidden reservoir e.g., in the spleen or bone marrow that cannot be detected by qPCR analysis of peripheral blood.

## Sporozoite-induced blood-stage infections with *app(−)* and *lap(−)* are resolved by mice

Next, we assessed the virulence of sporozoite-induced infections of the gene-deletion mutants *app(−)* and *lap(−)*, which have already shown full clearance in C57BL/6 mice after iRBC-induced infections and were able to produce infectious sporozoites after mosquito infection (Figs. 1C and 2; Table EV1 and Appendix Table S6). Infection of C57BL/6 mice by bite of infected mosquitoes or by intravenous injection of 1000 *app(−)* or *lap(−)* sporozoites or infection of SWISS mice by intravenous injection of 10,000 *app(−)* or *lap(−)* sporozoites resulted in blood-stage infections but with a significant delay in blood-stage patency ($P < 0.0001$, Dunnett´s multiple comparisons test) and a reduced blood-stage growth rate compared to mice infected with wild-type sporozoites (Table 1; Fig. 2D–I; Appendix Fig. S9). Sporozoite-induced wild-type infections resulted in death of all 16 mice (Fig. 2F,L; Table 1; Appendix Fig. S9). In contrast, out of 44 *app(−)* sporozoite-infected mice only one C57BL/6 mouse infected with *app(−)* sporozoites died (Fig. 2F,I; Table 1; Table EV2 and Appendix Fig. S9). All 69 mice infected with *lap(−)* sporozoites survived (Fig. 2F,I; Table 1; Table EV2 and Appendix Fig. S9). In mice

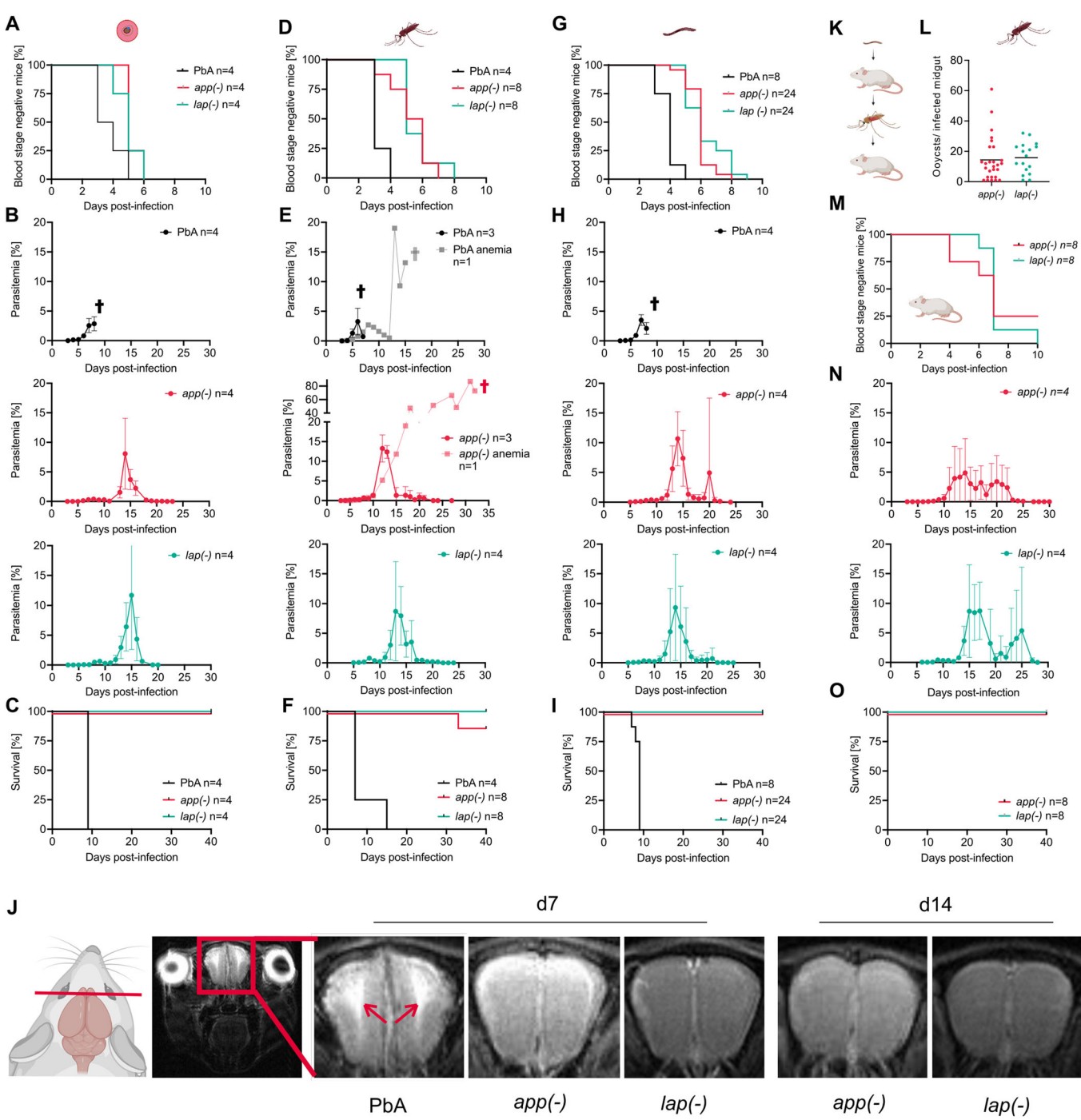

**Figure 2. Sporozoites lacking *app* and *lap* induce limited blood-stage infections.**

(A) Percentage of blood-stage-negative C57BL/6 mice infected with 100 wild-type, *app(−)*, and *lap(−)* blood-stage parasites (*n* = 4). (B, C) Mean parasitemia (B) and mouse survival (C) for the animals infected in (A). (D) Percentage of blood-stage negative C57BL/6 mice infected with wild-type (*n* = 4), *app(−)* (*n* = 8) and *lap(−)* (*n* = 8) sporozoites by ten biting mosquitoes. Mosquitoes were on average infected with 14,000 PbA, 6700 *app(−)* or 3800 *lap(−)* salivary gland sporozoites. (E, F) Mean parasitemia (E) of a subset of mice (*n* = 4) and mouse survival (F) for animals infected in (D). Light red parasitemia curve shows the infection course of the *app(−)* infected mouse that died. (G) Percentage of blood-stage-negative C57BL/6 mice infected with 1000 wild-type (*n* = 4), *app(−)* (*n* = 24) and *lap(−)* (*n* = 24) salivary gland-derived sporozoites. (H, I) Mean parasitemia (H, *n* = 4) of a subset of mice and mouse survival (I, *n* = 4 or *n* = 24) of animals infected in (G). (J) Representative MRI images showing brain edema at the level of the olfactory bulb (arrows) in mice infected by wild-type (*Pb*A) at 7 days post infection with 1000 sporozoites but not in mice infected with *app(-)* and *lap(-)* sporozoites. In total, four C57BL/6 were infected and imaged per parasite line. (K) Illustration of re-transmission experiment. (L) Oocyst prevalence of mosquitoes that were fed on mice 14 days after the mice were infected with *app(−)* and *lap(−)* sporozoites. (M) Prepatency of C57BL/6 mice (*n* = 8) infected by caged mosquitoes carrying re-transmitted *app(−)* and *lap(−)* parasites (mosquitoes from the experiment described in (L)). Six out of eight *app(−)* and all *lap(−)* infected mice became blood-stage patent. (N, O) Course of blood-stage infection (N, *n* = 4) and survival (O, *n* = 8) of mice infected in (M). (A–I, M–O) *n*: number of infected mice. (B, E, H, N) Mean parasitemia values ± standard deviation from four infected mice are shown. Source data are available online for this figure.

surviving from a sporozoite-induced infection, the infection peaked around day 14 to 17 and was ultimately resolved within 20–27 days (Table 1; Table EV1). The highest measured parasitemia was 70% (C57BL/6 infected with 1000 *app(−)* sporozoites), and the longest time to complete parasite clearance was 46 days with mice being infected for 39 days (C57BL/6 infected with 1000 *app(−)* sporozoites; Table EV1). Even at high parasitemias, all surviving *app(−)*- and *lap(−)*-infected mice showed no signs of disease. To evaluate possible signs of ECM, C57BL/6 mice were subjected to magnetic resonance imaging (MRI) on day 7 and 14 post-infection with 1000 salivary gland sporozoites to assess brain morphology and appearance of edema (Hoffmann et al, 2016). While mice infected with wild-type parasites showed clear signs of ECM already at low parasitemias at day 7 post,infection, no signs of ECM were detected in *app(−)*- or *lap(−)*-infected mice (Fig. 2J). Even when *app(−)* and *lap(−)*-infected mice showed high parasitemias around day 14 post,infection, no signs of edemas in the olfactory bulb were detectable.

In addition, we assessed infections of C57BL/6 with higher numbers of 1000 *lap(−)* iRBCs and 10,000 *lap(−)* sporozoites (Fig. EV1). These mice became blood-stage positive at day 5 and day 4, respectively, showed peak parasitemias of 11% and 21% at day 15 and day 12, respectively, and cleared the infection by day 23 and 20, respectively (Fig. EV1). Also, these mice showed no signs for ECM as measured by the rapid murine behavior and coma score and did not lose weight during the course of infection (Fig. EV1).

We next tested if sporozoite-induced *app(−)* and *lap(−)* blood-stage parasites could be transmitted back to mosquitoes. To this end we allowed mosquitoes to feed on infected mice prior to resolving the blood-stage infection (Fig. 2K,L). These mosquitoes were able to infect C57BL/6 mice 21 days post-infection demonstrating that these mutant parasites can be cycled between vector and host (Fig. 2M–O).

## Mice infected with *app(−)* and *lap(−)* parasites are protected from challenges with wild-type sporozoites

We next investigated if C57BL/6 mice that cleared an *app(−)* or *lap(−)* infection could survive a challenge with wild-type sporozoites. To this end we challenged mice that resolved an *app(−)* and *lap(−)* blood-stage- or sporozoite-induced infection with wild-type sporozoites at 90, 180, and 360 days post,infection (Fig. 3A–H; Table 2; Appendix Table S7). Around half of the challenged mice did not develop a detectable blood-stage infection after the challenge at 90 days (Fig. 3B; Table 2; Appendix Table S7). The other half developed a low-level blood-stage infection with the onset of all blood-stage infections delayed by 3–9 days compared to age-matched naive mice. The blood-stage infections peaked between 9 and 19 days at less than 2% parasitemia and were cleared after ~20 days (Fig. 3; Table 2; Appendix Fig. S10). When challenged at 180 days, 60% (*app(−)*) and 20% (*lap(-)*) of the mice did not develop a blood-stage infection. The mice that developed a blood-stage infection showed parasitemias of maximum 2.4% for *lap(−)* and the longest time to complete parasite clearance was 19 days with mice being infected for 13 days. Challenge after 360 days resulted in blood-stage infections of 4 out of 4 mice (*app(−)*, average peak parasitemia of 2.3%), and 1 out of 4 mice (*lap(−)*, peak parasitemia of 0.7%) (Table 2; Appendix Table S7). All challenged mice were able to resolve the wild-type infection and survived.

We next tested if we could mimic the course of infection followed by parasite clearance as observed with the *app(−)* and *lap(−)* parasites by curing wild-type-infected mice. To this end, we infected C57BL/6 mice with the NK65 strain, which allows for longer blood-stage infection leading to higher parasitemia and anemia (Akkaya et al, 2020) (Fig. 3I–K). We then treated these mice with chloroquine at day 16 post-infection which led to clearance of the infection and survival of all mice while untreated mice died after 25 days post-infection (Fig. 3J,K). Protective efficacy of this regimen was tested by challenging these mice with 1000 wild-type PbA sporozoites. All challenged mice became blood-stage positive between day 4 and 7, with control mice dying latest on day 10 post-challenge. In contrast, all chloroquine-treated mice survived, reaching peak parasitemias of 1.1–2.6% between days 7 and 11 and cleared the infection by days 11–12 (Fig. 3L–N; Appendix Table S8).

Finally, we tested if mice immunized with either *app(−)* or *lap(−)* *P. berghei* sporozoites that resolved infections were protected from heterologous challenge. To this end, we challenged the mice 90 days post-infection with 1000 *P. yoelii* sporozoites. Only half of the mice developed a *P. yoelii* blood-stage infection. In contrast to infected naive control mice, which all died, all *app(−)* or *lap(−)* immunized mice survived this heterologous challenge and cleared the infection, indicating cross-species immunity (Table 2). *P. yoelii*-induced infections appeared as fast in the blood as *P. berghei* in the *app(−)* immunized mice but showed higher peak parasitemias than in *P. berghei* challenges (Table 2). In contrast, in *lap(−)*-immunized mice challenged with *P. yoelii* sporozoites infections appeared 4 days earlier than in *lap(−)*-immunized mice challenged with *P. berghei* sporozoites (Table 2). Intriguingly, the *P. yoelii*-challenged mice showed a lower average peak parasitemia (0.2%) than *P. berghei*-challenged mice (0.4%).

Together, these results suggest that differences exist between *app(−)* and *lap(−)* parasites in inducing immunity by infection with sporozoites. Importantly, our data show that a single infection with low sporozoite numbers of gene-deletion mutants with reduced blood-stage growth and virulence can result in complete protection from a lethal challenge with wild-type parasites.

## Clearance of *lap(−)* blood-stages is mediated by antibodies, while protection from challenge infections depends on CD4+ T cells

The clearance profile during *app(−)* and *lap(−)* induced infections suggests that clearance is mediated by antibodies as the drop of infection starts around 2 weeks post infection. To investigate this hypothesis, we depleted B cells in mice by injecting either monoclonal antibodies targeting CD20 (Uchida et al, 2004) or control antibodies, followed by infection with 100 *lap(−)* iRBCs (Fig. 4A; Appendix Fig. S11A,B). Antibody treatment was repeated weekly during the course of infection. All mice became blood-stage positive between day 6 and 9 (Fig. 4B). While mice injected with control antibodies survived and cleared the infection similar to previous infections, all mice receiving the anti-CD20 antibodies died (Fig. 4C–E). Of these mice three succumbed early and one mouse died late (Fig. 4C,D). Serum of *lap(−)*-infected mice taken 30–40 days post infection readily detected blood-stage parasites by Western blot and immunofluorescence microscopy, indicating the presence of anti-*P. berghei* antibodies (Appendix Fig. S12).

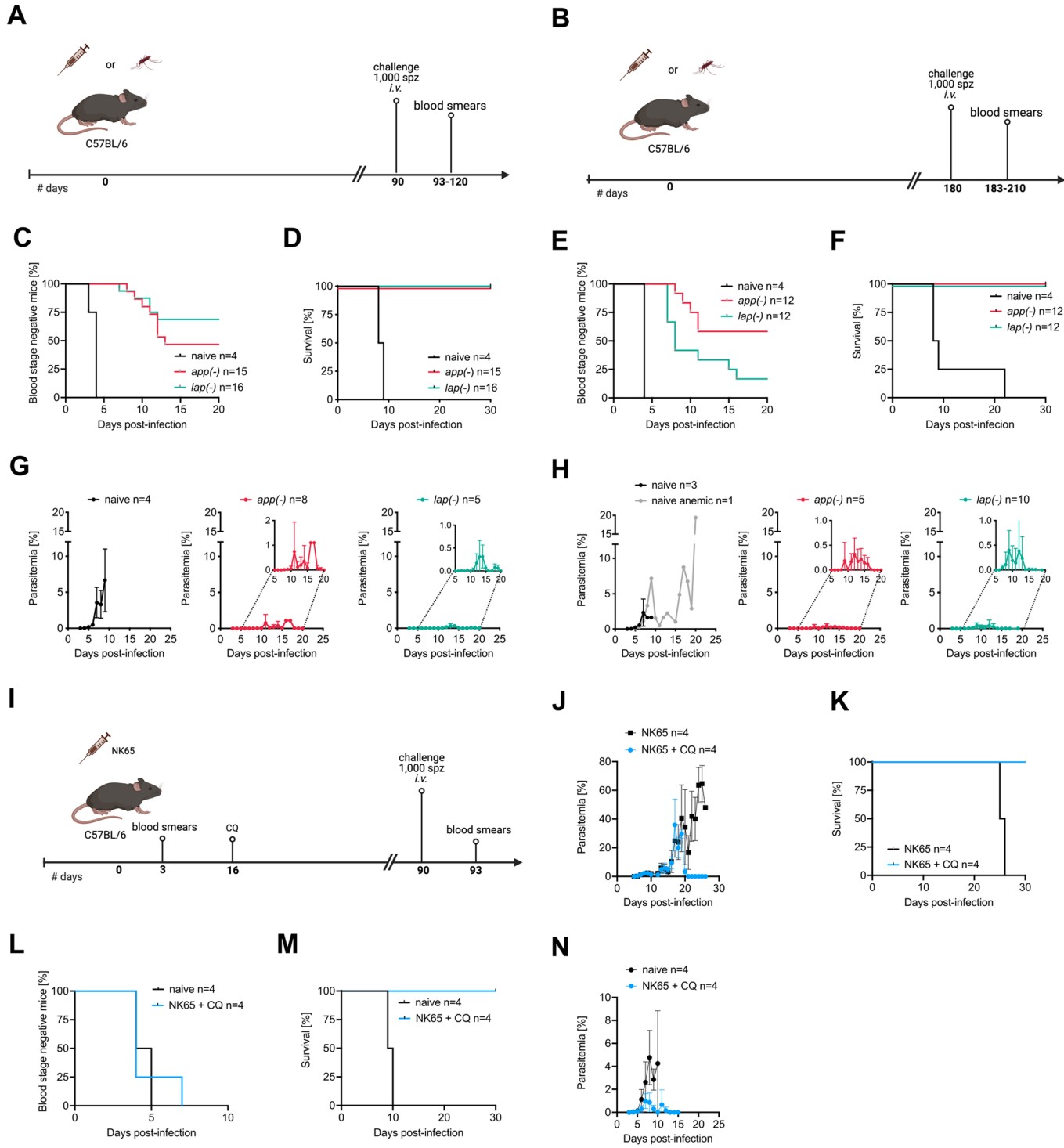

**Figure 3. Limiting infections cause protection from lethal wild-type challenge.**

(A, B) Illustration of challenge experiments. C57BL/6 mice were immunized with either 1000 salivary gland-derived sporozoites or by the natural transmission of *app(−)* or *lap(−)* parasites and challenged with 1000 salivary gland-derived wild-type sporozoites 90 (A) or 180 (B) days post-immunization. (C–F) Percentage of blood-stage negative (C, E) and surviving (D, F) mice after challenge with wild-type sporozoites after 90 (C, D) and 180 (E, F) days. (G, H) Mean parasitemia of naive (*n* = 4) or immunized C57BL/6 mice (*n* = 8 *app(−)* and *n* = 5 *lap(−)*) as challenged in (A) or (B), respectively. Small insets display magnifications of parasitemia curves. Mean parasitemia values ± standard deviations from all infected mice are shown. (I) Illustration of challenge experiment of *P. berghei* strain NK65 immunized mice. (J) Parasitemia of NK65 infected mice. C57BL/6 mice (*n* = 8) were infected with 100 NK65 iRBC. Four C57BL/6 were treated with chloroquine (CQ) starting on day 16 post-infection. Mean parasitemia values ± standard deviations from all infected mice are shown. (K) Survival of mice infected in (J). (L, M) Percentage of blood-stage negative (L) and surviving (M) C57BL/6 mice after challenge with 1000 PbA wild-type sporozoites. (N) Parasitemia of naive or NK65 immunized C57BL/6 as challenged in (L). Mean parasitemia values ± standard deviations from all infected mice are shown. Source data are available online for this figure.

**Table 2. Protection of C57BL/6 mice immunized with *app*(−) or *lap*(−) sporozoites against wild-type challenge 90 days, 180 days, or 360 days post-immunization.**

| Challenge with 1000 spz | | | Animals infected/animals challenged | Average prepatency [d] | Average peak parasitemia [%] | Animals dying/animals challenged |
|---|---|---|---|---|---|---|
| *P. berghei* | Naive control | d90 | 4/4 | 3.8 | 6 | 4/4 |
| | | d180 | 4/4 | 4 | 6.9 | 4/4 |
| | | d360 | 4/4 | 4.3 | 6.4 | 4/4 |
| | *app*(−) | d90 | 8/15 | 10.6 | 0.7 | 0/15 |
| | | d180 | 5/12 | 9.2 | 0.4 | 0/12 |
| | | d360[a] | 4/4 | 7.8 | 2.3 | 0/4 |
| | *lap*(−) | d90 | 5/16 | 10 | 0.4 | 0/16 |
| | | d180 | 10/12 | 9.4 | 0.6 | 0/12 |
| | | d360[a] | 1/4 | 6 | 0.7 | 0/4 |
| *P. yoelii* | Naive control | d90 | 5/5 | 4.8 | 27.7 | 5/5 |
| | *app*(−) | d90 | 3/8 | 12 | 1.4 | 0/8 |
| | *lap*(−) | d90 | 4/8 | 5.8 | 0.2 | 0/8 |

[a]Mice already challenged at d180 were rechallenged at d360.

To test mechanisms of protection induced by *lap*(−) infections, we investigated the role of CD4[+] and CD8[+] T cells. To this end, we immunized C57BL/6 mice with 1000 *lap*(−) sporozoites and treated them prior to a wild-type challenge infection with antibodies targeting either CD4[+] or CD8[+] T cells or control antibodies (Fig. 4F; Appendix Fig. S11A,C,D). Antibody treatment was repeated every five days after challenge, in total four times, and blood smears were analyzed throughout the course of infection. Mice treated with a control antibody showed a protection profile comparable to previous experiments, with around 50% of the mice being infected and clearing the infection (Figs. 3C and 4G,H). A similar infection course was observed for mice depleted of CD8[+] T cells (Fig. 4G). In contrast, all mice depleted of CD4[+] T cells became blood-stage patent with one mouse dying before the antibody treatment was stopped (Fig. 4H,I). Taken together, these experiments suggest that antibodies are key for clearance and CD4[+] T cells for protection.

As cytokines like IFNγ and IL-12 are important for antibody-mediated control of and protective immunity to different rodent-infecting *Plasmodium* parasite infections (Su and Stevenson, 2000, 2002; Postma et al, 1999; Angulo and Fresno, 2002), we further examined whether specific cytokines are elevated upon immunization with *lap*(−) parasites. To this end, we infected 4 C57BL/6 mice with 100 *lap*(−) iRBC, took serum 86 days post-infection and analyzed levels of a variety of cytokines. In mice immunized with *lap*(−) iRBC, the pro-inflammatory cytokines IFNγ, TNF-α, IL-1β, IL-12p70, IL-17A, MCP-1, and IL-23 were indeed significantly increased compared to naive mice (Fig. 5) and might therefore be involved in modulating antibody-mediated clearance of *lap*(−) infections and protection from wild-type challenge.

## Generation of slow-growing *P. falciparum* mutant parasites

To generate *P. falciparum* gene-deletion mutants with a slow blood-stage growth rate we used CRISPR/Cas9-based methods to delete (parts of) selected genes and introduce premature stop codons. In parallel, we generated control mutants in which, we introduced silent mutations into the selected/target genes (Fig. 6; Appendix Fig. S13A,B), performing in total over 50 independent transfections. We first targeted the *P. falciparum app* and *lap* genes for deletion. These two *P. falciparum* genes were previously reported to be refractory to gene deletion, indicating they are essential for in vitro blood-stage growth (Dalal and Klemba, 2007; Edgar et al, 2023). In line with these observations, we were not able to delete these *P. falciparum* genes in multiple transfection attempts (Appendix Table S9 and Appendix Fig. S13C,D). We next targeted an additional six genes based on evidence for the slow growth of gene-deletion mutants obtained in genome-wide knockout screens in *P. berghei* and *P. falciparum* and published reports (Appendix Table S9) (Balu et al, 2010; Zhang et al, 2018; Davies et al, 2020; Matthews et al, 2013; Matz et al, 2013). These six genes encode for *Plasmodium* exported protein (HYP11), thioredoxin 2 (TRX2), TIF eIF-2B delta subunit (EIF-2B), nicotinamide mono nucleotide adenylyl transferase (NMNAT), serine/threonine protein kinase FIKK family (FIKK8), and mago nashi protein homolog (Mago Nashi). We generated and selected gene-deletion mutants and matching control lines with silent mutations for all these six genes (Fig. 6; Appendix Fig. S13E–J). For two mutants, we observed a slow proliferation of their blood-stages in vitro at 80% and 70% of the control growth rate (Fig. 6A,B). These mutants either lack the gene encoding TRX2 or FIKK8, respectively. The other four mutants showed a wild-type-like asexual growth rate (Fig. 6A,B; Appendix Table S9). Analysis of *Pf trx2*(−) blood-stage development in synchronized cultures revealed a delay in the development of late trophozoites/early schizonts (Fig. 6C), in line with previously published data for *P. berghei trx2*(−) (Matthews et al, 2013; Matz et al, 2013). Overall, these results show that it is also possible to generate and select clones of slow-growing blood stages of *P. falciparum* gene-deletion mutants, despite the lower transfection efficacy and longer asexual cycle of *P. falciparum* compared to *P. berghei*. Importantly, the CRISPR-Cas9-based approach used in this

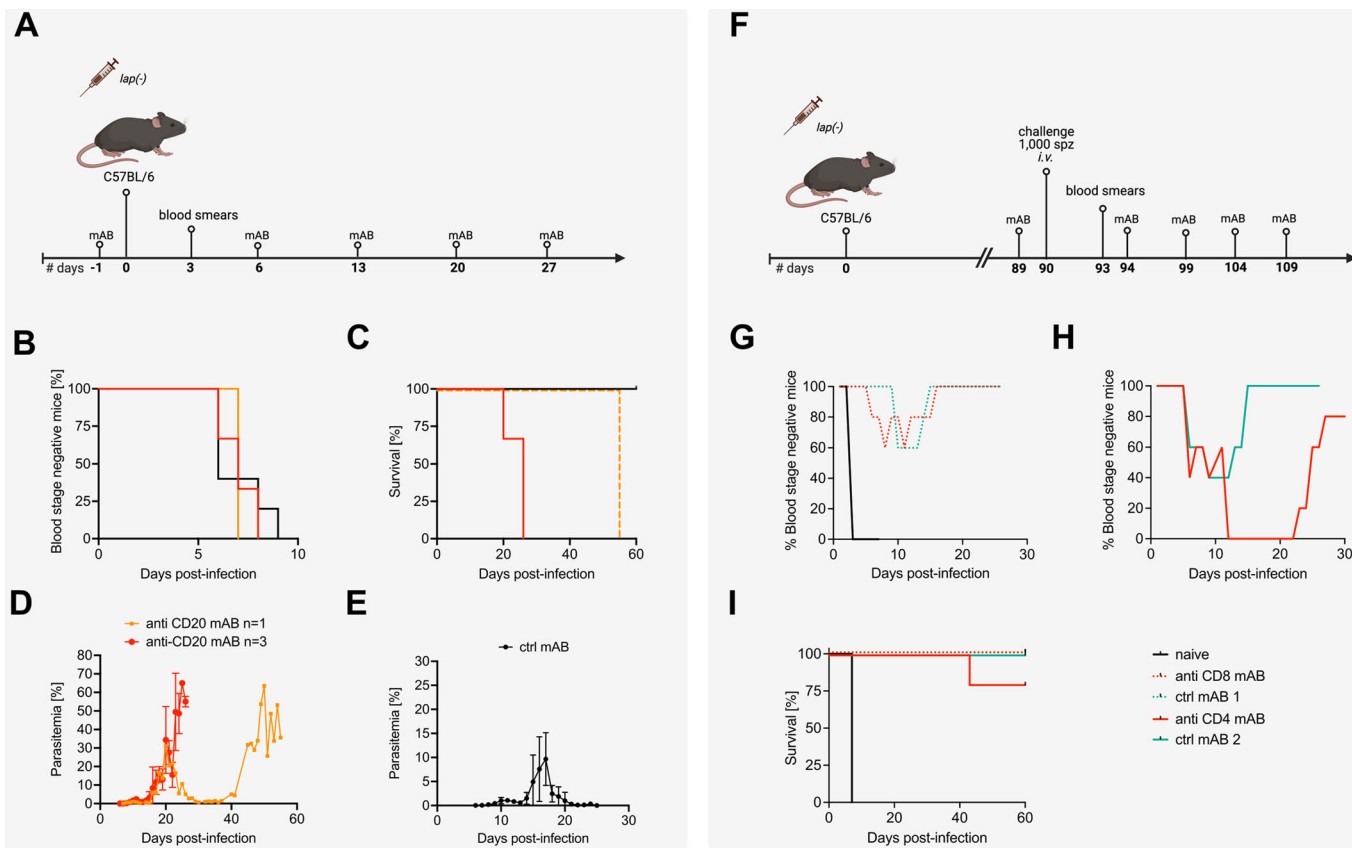

**Figure 4. Clearance of and protection by *lap(−)* parasites is mediated by B cells and CD4+ T cells, respectively.**

(A) Illustration of the B-cell depletion experiment. Four C57BL/6 mice each were treated with either anti-CD20 monoclonal antibody (mAB) or a control mAB one day prior to the infection with 100 *lap(−)* iRBC. Treatment with mAB was repeated every 7th day. (B) Percentage of blood-stage negative C57BL/6 mice infected with 100 *lap(−)* blood-stages. All mice became blood-stage positive between day 6 and day 9 post-infection. (C) Survival of all mice infected in (B). All B cell-depleted *lap(−)*-infected mice died between day 20 and 55 post-infection. All control mice survived the infection with *lap(−)* parasites. (D, E) Parasitemia of CD20-depleted or control *lap(−)*-infected C57BL/6 mice from (B). Mean parasitemia values ± standard deviations from all infected mice are shown. (F) Illustration of the CD4+ and CD8+ T cell depletion experiment. Twenty C57BL/6 mice were immunized with 100 *lap(−)* iRBC. One day prior to the challenge with 1000 wild-type sporozoites, five mice were treated with anti-CD4 or anti-CD8 mAB, and five mice were treated with the respective control mABs. Treatment with depletion antibodies was repeated every 5th day. Five C57BL/6 were used as naive control group. (G, H) Percentage of blood-stage negative mice after challenge with wild-type sporozoites. Protection against wild-type challenge did not differ between CD8+ T cell-depleted and control mAB-treated mice (G). CD4+ T cell-depleted mice were less protected against challenge than matching control mice (H). All naive mice became blood-stage positive and were not protected (G). (I) Survival of all mice infected in (G, H). All CD8+ T-cell-depleted mice and all control mice survived the challenge with wild-type parasites. In contrast one of the CD4+ T cell-depleted mice died from wild-type challenge on day 43 post-infection. All naive challenged mice succumbed to death on day 7 post-infection. iRBC infected red blood cells, mAB monoclonal antibodies, ctrl control. Source data are available online for this figure.

study does not result in the stable integration of foreign genetic material, such as drug-selectable markers or CRISPR/ Cas9 sequences, into the genome of *P. falciparum*. *P. falciparum* mutant parasites lacking any transgenes can thus be obtained by prolonged blood-stage cultures, which will result in the loss of the episomal plasmid. The absence of transgenic DNA is essential for the approval of future use of these mutant parasites as genetically attenuated Wbs in immunization approaches in humans.

## Discussion

Here, we generated and screened 18 new *P. berghei* gene-deletion mutants for attenuated blood-stage growth to use in experimental whole blood-stage (Wbs) and whole sporozoite (Wsp)

immunization approaches. We identified two mutants, *lap(−)* and *app(−)*, that could complete the parasite life cycle and showed slow blood-stage growth in mice both after infection with sporozoites and with blood-stage parasites. All 84 mice infected with either *lap(−)* blood stages or sporozoites resolved blood-stage infections within less than 4 weeks and remained parasite-free (analyzed by Giemsa-stained blood smears) for up to 180 days. This corresponds to a 95% binomial confidence interval of 0–4.4%. Following challenge with wild-type *P. berghei* or *P. yoelii* sporozoites, these mice were either sterile protected or cleared blood-stage infections without showing signs of disease. Cross-species protection has been observed in various studies using rodent malaria models with mixed results depending on parasite species, mouse strain and (co-)infections (Voza et al, 2005; Craig et al, 2012). Yet, these studies should be treated with caution in

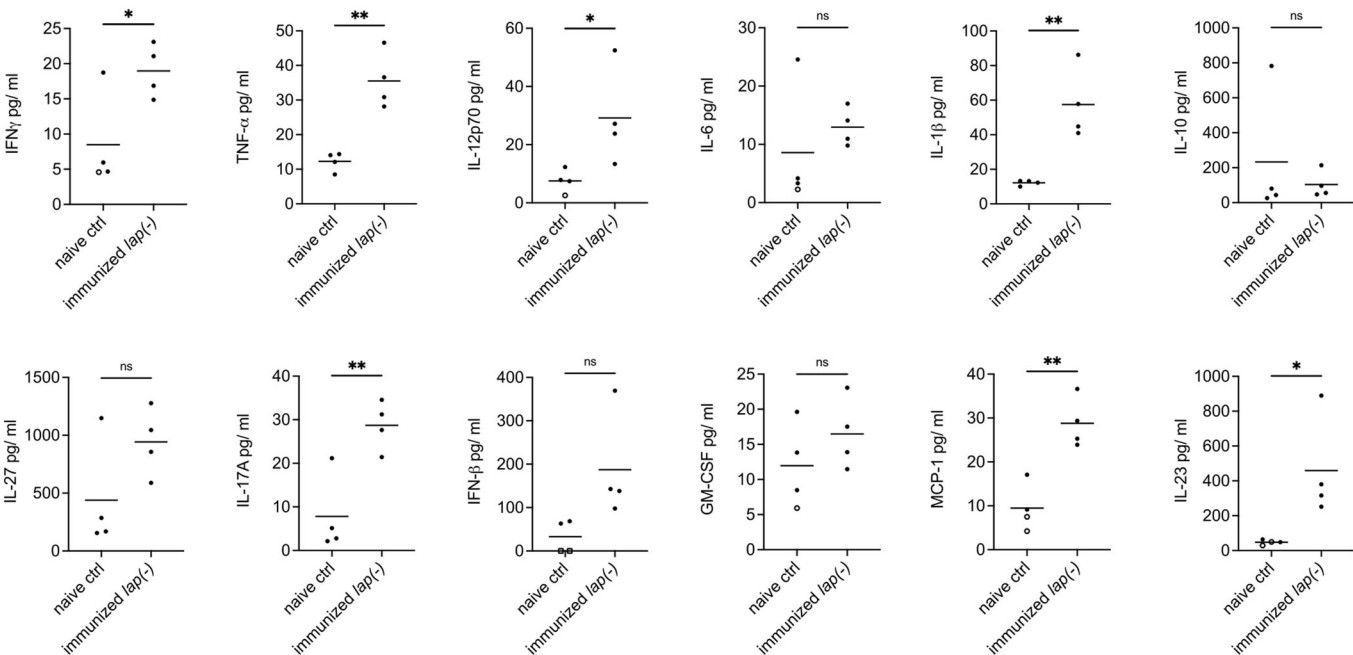

**Figure 5. Pro-inflammatory cytokines are increased upon immunization with *lap*(−) iRBC.**

Cytokine multiplex assay on sera of *lap*(−) immunized C57BL/6 (*n* = 4) showed significantly elevated levels of IFNγ, TNF-α, IL-12p70, IL-1β, IL-17A, MCP-1, and IL-23 compared to naive control mice (*n* = 4). The levels of the regulatory cytokine IL-10 were not significantly different between immunized and control mice. Unpaired *t* test, *<0.05, **P < 0.01. ctrl control. Filled circle, mean of two technical replicates; unfilled circle, one replicate was excluded as it was below threshold; unfilled square, both replicates were below threshold. Source data are available online for this figure.

regard to translation to human malaria (Langhorne et al, 2011). In our study, clearance went along with elevated cytokines and was likely mediated by antibodies. Protection from wild-type challenges induced by the attenuated parasites was dependent on CD4⁺ T cells, as was shown previously for other blood-stage attenuated parasites (Demarta-Gatsi et al, 2017; Aly et al, 2010). *App*(−) parasites showed similar characteristics although two out of 46 mice infected with blood-stages or sporozoites were not able to resolve the blood-stage infection and died. This higher pathogenicity of *app*(−) compared to *lap*(−) correlates with a more severe growth attenuation of *lap*(−) blood stages in vivo and in vitro. These subtle differences in protective efficacy warrant further studies.

Persistent infections of *P. berghei* in mice were shown to be necessary for their sustained immunity against challenge infections (Eling, 1980). While we cannot rule out sub-patent infections in all our mice, we examined a subset by qPCR and found that mice indeed cleared the infection. This result does not exclude a hidden reservoir, e.g., in the spleen from which new infections might recrudesce. This possibility will be important to consider for translational studies in *P. falciparum*, where sub-patent infections can occur for a long time (Portugal et al, 2016; O'Flaherty et al, 2022).

The generation of *P. falciparum* parasite lines with similar strongly reduced growth and virulence as for *P. berghei lap*(−) blood-stages will help to establish controlled human infections with longer exposure of human volunteers to blood-stage infections before clearance with antimalarials is necessary. This approach will permit the gain of novel insights into immune responses induced

by genetically attenuated Wbs in humans. However, we and others were not able to produce similar *P. falciparum* mutants lacking the gene encoding LAP using a CRISPR/Cas9 gene-deletion approach (Dalal and Klemba, 2007; Edgar et al, 2023), excluding further analysis on both growth and virulence features of such *P. falciparum* Wbs. We therefore continued to generate *P. falciparum* mutants with a slow growth rate by selecting target genes from previously published studies (Zhang et al, 2018; Bushell et al, 2017; Matthews et al, 2013; Matz et al, 2013). In a small-scale gene-deletion screen, we show that it is also possible to generate and select slow-growing blood-stages of *P. falciparum* gene-deletion mutants, despite the lower transfection efficacy and longer asexual cycle of *P. falciparum* (48 h) compared to *P. berghei* (24 h).

Recently, a genetically attenuated Wbs vaccine was tested with the candidate line lacking the knob-associated histidine-rich protein KAHRP, which is involved in the adhesion of infected red cells to the endothelium (Webster et al, 2021). This Wbs vaccine leads to increased splenic clearance of iRBCs and hence lowered virulence. However, high doses of parasites led to patent infections (Webster et al, 2021). Additional deletion in this *kahrp*(−) parasite line of a gene causing blood-stage growth attenuation as presented in our study might allow the use of higher parasite numbers for immunization and lead to longer avirulent infections in humans. In addition, combination of gene-deletions resulting in late liver-stage attenuated parasite lines (Kublin et al, 2017; van Schaijk et al, 2014; Roestenberg et al, 2020; Murphy et al, 2022; Goswami et al, 2024) with mosquito-transmittable blood-stage attenuated parasites could safeguard liver-stage attenuated parasite lines from virulent breakthrough infections.

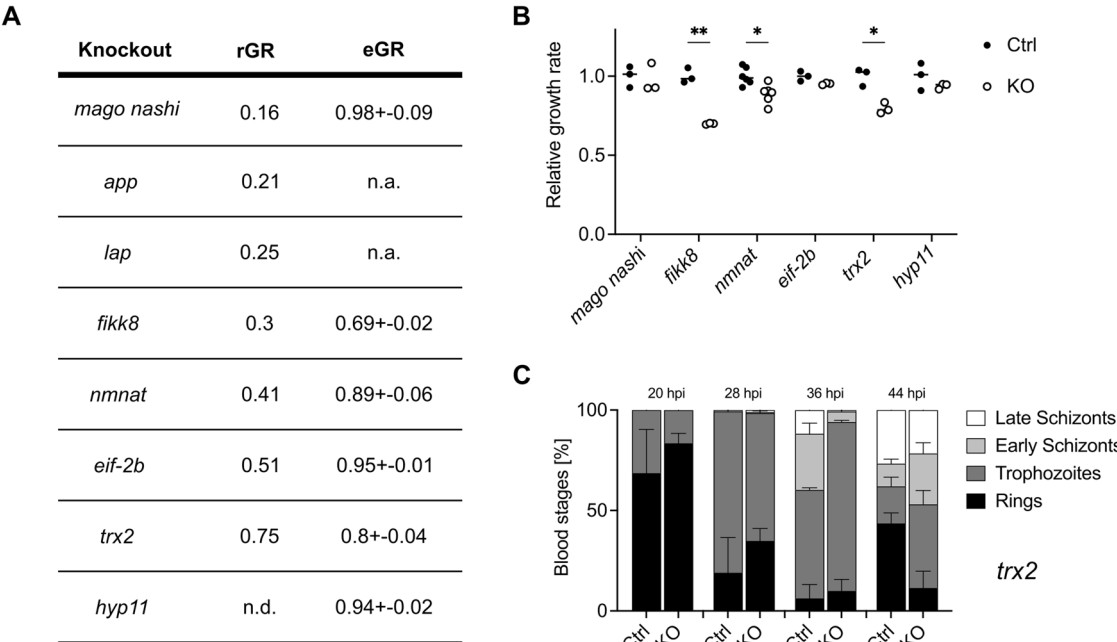

**Figure 6. Generation of slow-growing *P. falciparum* gene-deletion lines.**

(A, B) Predicted relative (rGR) (Zhang et al, 2018) and experimental (eGR; this study) growth rates of parasite lines; for gene IDs, see Appendix Table S9. Per knockout, experimental growth rates were normalized to the mean of the respective control line that was generated by the introduction of silent mutations. Unpaired *t* test with Holm–Šídák's correction for multiple comparisons, *$P < 0.05$, **$P < 0.01$. Three independent experiments were conducted except for *nmnat* where six independent experiments were conducted. (C) Stage composition of synchronized *Pf trx2*(−) and matching control lines. At least 100 parasites were staged per replicate and time point. Three independent experiments were conducted. rGR relative growth rate, eGR experimental growth rate, hpi hours post-infection, ctrl control, KO knockout. Mean percentage of parasite erythrocytic stage ± standard deviation is shown. Source data are available online for this figure.

In rodent malaria parasites, many other candidate genes could be investigated for their potential to cause self-limiting infections once ablated from the genome. Bushell et al identified 256 *P. berghei* genes that bestow a relative growth rate of 0.2–0.4 and another 133 with a relative growth rate of 0.4–0.6 upon their deletion (Bushell et al, 2017). As we were able to identify two candidates out of our list of 51 genes, simple interpolation would posit that from the remaining 338 genes around 13 more parasite lines with similar characteristics should be obtainable. However, growth rates of our individual gene-deletion mutants of both *P. berghei* and *P. falciparum* showed only a limited overlap with predicted growth rates of the genome-wide screens (Zhang et al, 2018; Bushell et al, 2017).

Our study also provides the basis for several additional strategies to generate attenuated blood-stage parasites with reduced growth and virulence that produce sporozoites and thus are transmissible. For example, when gene-deletion leads to reduced growth and/or virulence of blood-stages but in addition blocks further development in the mosquito, promoter swap strategies could be employed to express the gene during mosquito development by expressing the respective gene from a gametocyte or mosquito stage-specific promoter (Deligianni et al, 2011; Laurentino et al, 2011; Kehrer et al, 2016; Klug et al, 2018). In addition, mutant parasites with slow-growing blood stages that still show virulence characteristics could be used as a basis to generate double or triple gene-deletion mutants that might attenuate virulence. Finally, the human-infecting zoonotic *P. knowlesi* parasite could also be tested for

slow growth as an intermediate preclinical model (Chakravarty et al, 2022; Grüring et al, 2014) as it can be cultured in vitro like *P. falciparum* but shows a higher transfection efficacy and a faster blood-stage cycle of just 24 h (Mohring et al, 2020). Lastly, slow-growing *P. falciparum* parasites were recently also identified by a screen using selection-linked resistance marker integration, which could help to prescreen for gene deletions resulting in an avirulent phenotype (Kimmel et al, 2023).

In conclusion, we generated and screened a number of parasite gene-deletion mutants to obtain blood-stage attenuated malaria parasites in *P. berghei* and *P. falciparum*. We generated mutant *P. berghei* lines with reduced growth of blood-stages, and infection of mice by a low number of infected mosquito bites or by single injection of sporozoites of these mutants led to protection from disease after challenge with wild-type sporozoites. These new types of Wbs-attenuation will constitute valuable tools to inform on the induction of immune responses and may aid in developing as well as safeguarding attenuated parasite vaccines.

## Methods

### Bioinformatic identification of slow-growing parasites

Gene-deletion mutants showing a growth rate of 0.2–0.6 were retrieved from PlasmoGEM (https://plasmogem.umu.se/pbgem/). In addition, from this list of genes/mutants, genes were selected either

manually or using PlasmoSPOT (https://frischknechtlab.shinyapps.io/plasmoSPOT/) (Farr et al, 2021) for a transcription profile that shows transcription in asexual blood stages but low or no expression in gametocytes and mosquito stages.

## Ethics statement

All experiments were performed in accordance with GV-SOLAS and FELASA guidelines and have been approved by the German authorities (Regierungspräsidium Karlsruhe; G283/14, G-8/21, G9/21).

## Mice, parasites, and mosquitoes

SWISS mice were obtained from Janvier Laboratories (minimum 20 g) and C57BL/6 from Charles River laboratories (minimum 18 g). *Plasmodium berghei* ANKA, *Plasmodium berghei* NK65 and *P. yoelii* 17XL were used as indicated in the respective experiments. *Anopheles stephensi* was used for all experiments.

## Generation of *P. berghei* gene-deletion mutants

Gene-deletion DNA constructs containing the hDHFR/yFCU selectable marker cassette, allowing selection of transgenic parasites by pyrimethamine treatment of mice, were obtained from PlasmoGem (Gomes et al, 2015). Transfections were performed as described previously using synchronized *P. berghei* ANKA schizonts cultured overnight in RPMI-1640 complete (RPMI-1640 (GIBCO) supplemented with 20% FBS (US origin GIBCO) and 0.03% Gentamycin (10 mg/ml; PAA, Pasching, Austria)) at 37 °C, 5% $CO_2$ (Janse et al, 2006). Briefly, the purification of cultured schizonts was performed through a 55% Nycodenz/PBS gradient (Axis-shield diagnostics, Heidelberg, Germany). In total, 10 µg plasmid was digested with NotI overnight at 37 °C followed by ethanol precipitation and resuspension in 20 µl PBS. The linearized construct and the purified schizonts were mixed with Nucleofector solution from an Amaxa human T-cell Nucleofector Kit and electroporated using the Amaxa Nucleofector II device (Lonza, Köln, Germany). Directly after transfection 50 µl RPMI-1640 complete was added to the transfection reaction followed by intravenous injection into one SWISS mouse. Twenty-four hours post-transfection, drinking water was exchanged to 280 µM pyrimethamine (Sigma-Aldrich, Munich, Germany) to select for transgenic parasites. Parasitemia was assessed from day 7 onwards through Giemsa-stained thin blood smears. Once parasitemia reached 1.5% the blood was collected through cardiac puncture using a 1 ml syringe containing 50 µl heparin and mixed for storage in liquid nitrogen as follows: 100 µl blood, 200 µl Freezing solution (Alsever solution (Sigma-Aldrich, Munich, Germany) + 10% glycerol). In addition, genomic DNA was isolated from iRBCs by saponin-lysis followed by using the DNeasy Blood & Tissue Kit (Qiagen). Upon confirmation of correct integration by PCR isogenic parasite lines were generated by limiting dilution in mice (Waters et al, 1997). Only mutant lines with correct integration of the gene-deletion constructs and absence of the wild-type gene were used for further analysis. Primers used for genotyping PCRs are shown in Table S3.

## Blood-stage and mosquito infection

Blood was taken by cardiac puncture from a donor mouse at 0.2–1% parasitemia and upon serial dilution in PBS 100 infected red blood cells (per 100 µl PBS) were injected intravenously. Prepatency as well as parasitemia were assessed from day three post infection.

For mosquito infections, blood was taken by cardiac puncture from a donor mouse at 1–3% parasitemia. Recipient SWISS mice were treated with phenylhydrazine (0.6% in PBS, Sigma-Aldrich, Munich, Germany) to induce reticulocytes three days prior to being infected with 20 million iRBCs. Three days after infection, exflagellation was assessed and starved mosquitoes were allowed to bite anesthetized mice for 20 min.

## Evaluating mosquito infections

Presence of midgut oocysts was evaluated 10–14 days post-infection by first permeabilizing midguts with 0.01% NP40/PBS for 30 min followed by staining with 0.1% mercurochrome/$H_2O$ for 1 h. Following three washing steps with PBS, stained midguts were mounted on a glass slide, covered with a coverslip and assessed using a Zeiss Axiovert 200 M inverted microscope with a ×10 objective.

Number of sporozoites was assessed 18 days post-infection by isolating midguts and salivary glands of 10 mosquitoes. Sporozoites were released into RPMI-1640 by mechanically disrupting the tissue within 1.5-ml reaction tubes using a pestle and counted in a Neubauer chamber on a Zeiss Axiostar light microscope under a ×40 phase contrast objective.

In vitro motility assays were performed essentially as described before (Bane et al, 2016) with salivary gland-derived sporozoites isolated between days 18 and 24 post infection and purified with an Accudenz density gradient (Kennedy et al, 2012). Imaging was performed on a Zeiss Axiovert 200 M fluorescent microscope (25x objective, DIC, frame rate of 1/3 s for 3 min). FIJI (Schindelin et al, 2012) was used for the analysis of motility patterns as well as for the calculation of sporozoite speed.

## In vivo infections

For natural transmission experiments, 10 mosquitoes were transferred to a small cup 1 day before the experiment and starved overnight. The next day C57BL/6 were anesthetized and placed on top of the cups. Mosquitoes were allowed to feed for 10 min. Alternatively, sporozoites were isolated 18–23 days post-infection from mosquito salivary glands, sporozoites released, counted and injected into SWISS (10,000 sporozoites) or C57BL/6 (1000 sporozoites) mice by intravenous inoculation into the tail vein. Prepatency and course of parasitemia was assessed from day 3 post-infection onwards by Giemsa-stained smears from peripheral blood.

## Magnetic resonance imaging

C57BL/6 were infected by intravenous injection of 1000 salivary gland-derived sporozoites and parasitemia was monitored by Giemsa-stained blood smears starting day 3 post-infection. From day 6 post-infection, MRI of mice was performed with a 9.4-T horizontal bore small-animal MRI unit (BioSpec 94/20 USR; Bruker BioSpin, Ettlingen, Germany) with a four-channel phased-array surface receiver coil as described before (Hoffmann et al, 2017). Mice were anesthetized with 2% isoflurane. Anesthesia was maintained with 0.5–1.5% isoflurane. Animals were kept on a

heating pad to keep the body temperature constant, and respiration was monitored externally during imaging with a breathing surface pad controlled by an in-house–developed LabView program (National Instruments, Munich, Germany). To exclude the earliest manifestation of experimental malaria, the olfactory bulb was imaged with a 2D T2-weighted sequence (TR/TE = 2000/22 ms, slices = 12, and slice thickness = 0.7 mm).

## Challenge infections

Mice immunized with either 100 iRBC, 1000 infectious sporozoites or by mosquito bite were rechallenged 90 days, 180 days or 360 days after infection using 1000 freshly isolated PbA wild-type sporozoites. Appearance of blood-stage parasites was monitored daily starting three days post challenge by Giemsa-stained blood smears for at least 20 days. Mice challenged at 360 days received one previous challenge infection at d180.

## In vitro *P. berghei* hepatocyte invasion and exo-erythrocytic development

Hepatocyte invasion and exo-erythrocytic development were performed in HepG2 cells cultivated at 37 °C, 5% $CO_2$ in DMEM complete (10% FCS, 1% Penicillin/Streptomycin; Gibco). Two days prior to infection with 10,000 sporozoites, 100.000 (invasion) or 30.000 cells (exo-erythrocytic development) were seeded per well of an eight-well Permanox Lab-Tek chamber slide (Nunc). After two hours of incubation with sporozoites at 37 °C, 5% $CO_2$ wells were washed twice with DMEM complete and the invasion assay was stopped by adding 4% PFA for 20 min at RT while fresh DMEM complete supplemented with an Antibiotic–Antimycotic cocktail (1×, Gibco) was added to the other wells. After 48 h incubation at 37 °C, 5% $CO_2$ cells were fixed by adding ice-cold methanol for 10 min at RT. Both assays were washed twice with 1% FCS/PBS and then blocked for 2 h at 37 °C or o/n 4 °C in 10% FCS/PBS.

Cells in the invasion assay were stained subsequently for 1 h at 37 °C with primary mouse αCSP antibodies (cell culture super-natant 1:300 diluted in 10% FCS/PBS) and anti-mouse Alexa Fluor 488 antibodies (1:300 diluted in 10% FCS/PBS; Life technologies). Following two washing steps with 1% FCS/PBS the cells were permeabilized by addition of ice-cold methanol for 10 min at RT. Subsequently, another round of blocking with 10% FBS/PBS at 37 °C or at 4 °C overnight was followed by a second round of subsequent incubation with αCSP (cell culture supernatant 1:300 diluted in 10% FCS/PBS) and secondary anti-mouse Alexa Fluor 546 antibodies (1:300 diluted in 10% FCS/PBS; Life Technologies) for 1 h at 37 °C. Hoechst 33343 was added (1 µg/ml; Life Technologies) for 5 min at RT to visualize DNA. After two washing steps using 1% FCS/PBS the assays were mounted.

Cells in the exo-erythrocytic development assays were subsequently stained with a primary antibody against *Plasmodium berghei* HSP70 (cell culture supernatant 1:300 diluted in 10% FCS/PBS) and an anti-mouse Alexa Fluor 488 (1:300 diluted in 10% FCS/PBS; Life technologies). Hoechst 33343 was added after for 5 min at RT and the cells washed twice in 1% FCS/PBS the assays before being mounted. Cells were assessed at a Zeiss Axiovert 200 M inverted microscope and Fiji was used to measure sizes of liver-stages.

## In vitro *P. berghei* blood-stage development

Blood of infected mice was collected after mosquito infection, transferred to a 75-cm² cell culture flask containing RPMI-1640 supplemented with 20% FCS and 0.03% gentamycin, and cultivated at 37 °C, 5% $CO_2$. Sixteen hours later, schizonts were purified using a 55% Nycodenz gradient and centrifugation at 209 × g for 25 min without break. Schizonts were collected from the interphase and washed once with RPMI-1640 supplemented with 20% FCS and 0.03% gentamycin. Purified schizonts were intravenously injected into a naive SWISS. Two hours after infection blood was collected via cardiac puncture and transferred to a 75-cm² cell culture flask containing RPMI-1640 supplemented with 20% FCS and 0.03% gentamycin and cultivated at 37 °C, 5% $CO_2$. Starting 24 h post-infection and then every two hours 1 ml of culture was collected, and development of blood-stages was evaluated via Giemsa-stained blood smears. Developmental stages were classified as trophozoite (one nucleus), early schizont (multiple nuclei visible) or late schizont (single merozoites visible).

## qPCR for detection of blood-stages

In total, 10 µl tail blood was diluted in 500 µl PBS followed by centrifugation at 2000 rpm for 5 min in a tabletop centrifuge. The RBCs were lysed by 0.1% saponin and centrifuged at 14,000 rpm for 5 min in a tabletop centrifuge. Following one washing step with PBS, the lysate pellet was resuspended in 20 µl ddH₂O. To detect parasite-specific 18 S rRNA, qPCR using a 2× SBYR green Mastermix (Applied Biosystems) was performed in duplicates on 5 µl isolated DNA each on a Quantstudio 3 (Applied Biosystems). Lysate from blood of an uninfected mouse was used as a negative control.

## IFA on blood-stages

For the detection of blood-stages with parasite-specific IgG serum of immunized mice blood was harvested by cardiac puncture without heparin. The next day, the serum was collected and frozen. One mouse was infected by injection of frozen parasites (PbA 473 constitutively expressing RFP, generous gift of Kathie Hughes and Andy Waters, University of Glasgow). When parasitemia reached 2% blood was harvested by cardiac puncture with heparin. Red blood cells (RBC) were washed three times in PBS and then fixed with 4% PFA/0.0075% Glutaraldehyde for 30 min at 37 °C. After washing twice with PBS, sampes were permeabilized for using 125 mM Glycine/0.1% Triton-X-100 in PBS for 15 min at room temperature. Samples were blocked in 3% BSA/PBS for 2 h at room temperature followed by incubation of one sample with serum (1:500) overnight at 4 °C. The next day samples were washed three times 10 min each in PBS and then incubated with 1:300 anti-mouse Alexa Fluor 488 antibody for 1 h at room temperature. Following three washing steps with PBS samples were analyzed on a Zeiss Axiovert microscope. Hoechst-33342 (10 µg/ml) staining was performed during the second washing step. All sample incubations were performed rotating.

## Western blot

Blood from a highly PbA-infected mouse was lysed in RIPA buffer and $5 \times 10^6$ iRBC were mixed with SDS-sample buffer. WB samples were heated to 95 °C for 5 min and quickly spun down. After

separation on a 4–15% gradient Mini-Protean TGXTM Gel (Bio-Rad) proteins were transferred onto a 0.2 µm nitrocellulose membrane (Trans-Blot Turbo, Bio-Rad). 5% milk powder in TBST was used to block the membrane followed by cutting the membrane into several parts, each containing one sample of parasite lysate. Each part was individually incubated with 1:500 of the indicated sera in blocking solution overnight at 4 °C. The next day the membranes were washed 3 times in TBS before they were incubated with 1:10,000 secondary anti-mouse HRP (Roche) antibody in blocking solution for 1 h at room temperature. For detection, Pierce SuperSignal West Pico chemiluminescent substrate was used and imaging was performed on a Chemostar ECL & Fluorescence Imager (Intas). Afterwards, membranes were stripped from bound antibodies by incubation with 2% SDS/62 mM Tris pH 6.8/100 mM beta-mercaptoethanol in $H_2O$ for 30 min at 60 °C followed by three extensive washings in TBST for 10 min. The membrane was blocked again in 5% milk/TBST for 2 h at room temperature. Incubation with anti-$Pb$HSP70 (monoclonal IgG, 1:500 dilution) was performed overnight at 4 °C rotating. The next day the membranes were washed three times in TBST 10 min each at room temperature. Following incubation with secondary anti-mouse IgG-HRP (1:10,000 dilution, Roche) in 5% milk/TBST for 1 h at room temperature, the membranes were washed again three times in TBST for 10 min each before imaging was performed as described before.

### In vivo cell depletion

To investigate whether B cells are responsible for clearance C57BL/6 mice were treated with anti-CD20 antibodies (250 µg per mouse i.p., InVivoMAb clone MB20-11, Biolegend) one day before infection with 100 iRBC of Δ$lap$ and then every 7th day. As a control another group was treated with an isotype control antibody (250 µg InVivoMAb DV5-1 per mouse i.p., Biolegend). Whether $CD4^+$ T cells or $CD8^+$ T cells are important for the protection of $lap(-)$ immunized mice was tested by cell-specific depletion of either T cell population. One day prior wild-type challenge antibodies were injected $i.p.$ (300 µg InVivoMAb clone GK1.5 (Biolegend) per mouse and 200 µg InVivoMAb clone 53–5.8 (Biolegend) per mouse, respectively). Control groups of animals were treated with isotype control antibodies (300 µg InVivoMAb LTF-2 or InVivoMAb HRPN per mouse, respectively). Treatment with antibodies was repeated every 5th day, and depletion was confirmed by flow cytometry of 10 µl blood from the tip of the mouse tail. Blood was diluted in 500 µl ACK buffer, incubated for 3 min and then centrifuged for 5 min at 3500 rpm in a tabletop centrifuge. Antibodies against CD19 (1:100 anti-mouse CD19-A488 (clone 6D5), Biolegend), CD3 (1:100 anti-mouse CD3-PE (clone 145-2C11), Biolegend) and CD8 (1:100 anti-mouse CD8a-APC (clone 53-6.7), Biolegend) were used to stain for B cells, CD4 T cells and CD8 T cells. After incubation for 30 min at 4 °C in 0.5% BSA + 2 mM EDTA/PBS, cells were washed once in 0.5% BSA + 2 mM EDTA/PBS followed by fixation in 4% PFA/PBS for 10 min at RT. Cells were stored in PBS until analysis was performed on a FACS Celesta.

### Analysis of cytokine levels during immunization

C57BL/6 were immunized through intravenous injection of 100 $lap(-)$ iRBC. Parasitemia was monitored daily by Giemsa-stained blood smears. Eighty-six days post-infection, 100 µl blood were collected via the facial vein. The serum was stored at −20 °C until cytokine bead assay (LEGENDplex mouse inflammation panel (13-plex), Biolegend) was performed according to the manufacturer's instructions.

### NK65 + chloroquine

C57BL/6 mice were infected with 100 iRBC of NK65 and progression of parasitemia was monitored via Giemsa-stained blood smears. On day 16 post-infection, one group of mice was treated with 0.288 mg Chloroquine supplemented with 15 mg glucose/ml drinking water for 7 days. Wild-type challenge with 1000 $Pb$A sporozoites $i.v.$ was performed 90 days post-infection and parasitemia was monitored via Giemsa-stained blood smears.

### Cloning of constructs for the generation of *P. falciparum* knockout lines

To generate genetic deletion lines of target genes, we used a CRISPR/Cas9-based genetic strategy resulting in the loss of several hundred base pairs of gene sequence as well as the introduction of a premature stop codon and a frame shift (Appendix Fig. S13A). In parallel, control lines using the same gRNA only introduced a silent shield mutation into the target gene (Appendix Fig. S13B). Plasmids were based on pDC2-cam-coCas9-U6-hDHFR (a gift from M. Lee (Lim et al, 2016)). We designed gRNAs using the Eukaryotic Pathogen CRISPR gRNA Design Tool (Peng and Tarleton, 2015) and ordered forward and reverse sequences as unmodified single-stranded oligonucleotides (Merck). After annealing of the forward and reverse oligonucleotide, these were ligated into the BbsI-digested pDC2-cam-coCas9-U6-hDHFR plasmid. Homology regions were amplified by PCR using the primers indicated in Appendix Table S3 and cloned into the EcoRI/AatII-digested vector by Gibson assembly (NEBuilder HiFi DNA Assembly, New England Biolabs GmbH). Plasmid sequences were verified by control digest and sequencing. For transfection, medium-scale plasmid preparations were produced using the NucleoBond Xtra Midi Kit (Macherey-Nagel).

### *Plasmodium falciparum* culture and transfection

*P. falciparum* 3D7 parasites were routinely cultured in human 0+ erythrocytes at 5% hematocrit in complete RPMI-1640 medium and maintained at 37 °C under 3% $CO_2$, 5% $O_2$. Complete RPMI medium consisted of RPMI-1640 (GlutaMAX, Gibco, Thermo Fisher Scientific) supplemented with 0.5% AlboMAX II Lipid-Rich BSA (Gibco, Thermo Fisher Scientific), 0.2 mM hypoxanthine (c.c. pro-GmbH), 25 mM Hepes (pH 7.3; Sigma-Aldrich, Merck) and 12.5 µg/mL gentamycin (Carl Roth). Parasitemia was monitored using Giemsa-stained blood smears and parasites were maintained at parasitemias below 2%. If required, parasites were synchronized using standard sorbitol synchronization. Briefly, packed blood cells were incubated in 10 volumes of 5% sorbitol for 20 min at 37 °C, then pelleted and washed once with RPMI complete before return to culture.

For transfection, 50–100 µg purified and ethanol-precipitated DNA were resuspended in 30 µl TE buffer and 370 µl CytoMix and mixed with 200 µl packed erythrocytes containing at least 4% ring-stages. Parasites were electroporated (310 kV, 950 µF) using a Gene

**The paper explained**

**Problem**

A highly efficient malaria vaccine remains elusive. Currently, two vaccines are approved that are based on a single parasite surface protein. As alternatives to such protein-based vaccines, the use of attenuated live *Plasmodium* parasites is also explored. These can be either attenuated mosquito-derived sporozoites, which arrest their development in the liver or attenuated blood-stage parasites. Both approaches would profit from the availability of transgenic avirulent parasites that show a reduced multiplication rate in the disease causing blood-stage.

**Results**

Here we propose and test a new strategy for the generation of genetically attenuated parasites. We aimed to delete genes such that blood-stage parasites grow slow but still develop normally in the mosquito. Screening a large number of mutant rodent-infecting parasite lines, we discovered two mutants lacking genes important for hemoglobin digestion that caused mild and limited infections in mice while retaining full transmissibility through mosquitoes. These mice remained protected from disease after challenge with wild-type sporozoites in a manner depending on CD4+ T cells. Finally, we were able to generate transgenic slow-growing human-infecting *P. falciparum* lines.

**Impact**

Slow-growing, avirulent *P. falciparum* mutants will constitute valuable tools to inform on the induction of immune responses and will aid in developing new as well as safeguarding existing attenuated parasite vaccines.

Pulser II (Bio-Rad), and transgenic parasites were selected for by drug selection using 2.5 μM WR99210 (Jacobus Pharmaceutical Company). For genotyping transgenic parasite lines, genomic DNA was isolated from saponin-lysed and washed parasite pellets using the DNeasy Blood & Tissue Kit (Qiagen). PCR products from the genotyping reaction were sent for sequencing.

### Growth assay

To measure parasite growth, sorbitol-synchronized ring-stage parasites of mutant lines and their matching controls were seeded at a parasitemia of 0.05% and 4% hematocrit into 6-well dishes. Medium was changed daily and smears and samples for flow cytometry (100 μl culture) were taken every 24 h for three consecutive asexual cycles (6 days).

To monitor stage development, parasites were tightly synchronized to a 4 h window. To this end, schizonts were purified by underlaying the culture with 65% Nycodenz solution (Axis-shield diagnostics) in RPMI and centrifuged for 25 min at 1000 rpm without brake. The interphase containing schizonts was collected and washed once, and the parasites were mixed with medium containing blood at 4% hematocrit, transferred to a six-well plate, and cultured for 4 h to allow for reinvasion, followed by a round of sorbitol synchronization. Parasites were followed for a full cycle, collecting blood smears and samples for flow cytometry (100 μl culture) every 8 h.

Cell samples for flow cytometry were pelleted, fixed in 200 μl 4% PFA/0.0075% glutaraldehyde/PBS overnight at 4 °C and then stored in PBS. For quantification of parasitemia, cells were stained with SybrGreen (1:1000; Thermo Fisher Scientific) in PBS for 15 min at 37 °C, washed once in 200 μl PBS and analyzed on a BD FACS Celesta. Parasitemia was determined as the percentage of

SybrGreen-positive cells of all single cells. Growth rates were calculated as the average increase in parasitemia in two consecutive asexual cycles and normalized to the average growth rate of matching control parasite lines.

### Statistical analysis

All statistical analysis was undertaken using GraphPad Prism 9.2.0 (GraphPad, San Diego, CA, USA). Before statistical analysis data were analyzed for normal distribution, and the appropriate statistical tests were chosen. Animal infections were performed in groups of four unless stated otherwise. Analysis of experiments was not blinded. All exact *P* values indicated in the manuscript can be found in Appendix Table S10.

## Data availability

This study includes no data deposited in external repositories.

The source data of this paper are collected in the following database record: biostudies:S-SCDT-10_1038-S44321-024-00101-6.

## Peer review information

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

## Acknowledgements

The authors thank the PlasmoGEM consortium for the continuous and generous supply of the transfection vectors; Miriam Reinig for mosquito breeding; Steffen Borrmann, Elias Farr, Axel Roers, Katharina Morath, Oliver Fackler, and Mirko Singer for helpful discussions and suggestions on experiments and manuscript text. The authors thank specifically the late Shahid Khan for motivating and lively discussions. Figures were created partially with BioRender.com. The work was funded by grants from the German Center for Infection Research (DZIF, TTU 03.813), the German Research Foundation (FR2140/6-1; FR2140/10-1 and SFB 1129 project ID 240245660), and a Volkswagen Foundation *Experiment!* grant (Az 92393).

## Author contributions

**Julia M Sattler**: Conceptualization; Supervision; Investigation; Visualization; Writing—original draft; Project administration; Writing—review and editing. **Lukas Keiber**: Investigation; Writing—review and editing. **Aiman Abdelrahim**: Investigation; Writing—review and editing. **Xinyu Zheng**: Investigation; Writing—review and editing. **Martin Jäcklin**: Investigation; Writing—review and editing. **Luisa Zechel**: Investigation; Writing—review and editing. **Catherine A Moreau**: Investigation; Writing—review and editing. **Smilla Steinbrück**: Investigation; Writing—review and editing. **Manuel Fischer**: Methodology. **Chris J Janse**: Investigation; Writing—original draft; Writing—review and editing. **Angelika Hoffmann**: Supervision; Writing—review and editing. **Franziska Hentzschel**: Conceptualization; Supervision; Investigation; Visualization; Writing—review and editing. **Friedrich Frischknecht**: Conceptualization; Funding acquisition; Visualization; Writing—original draft; Project administration; Writing—review and editing.

Source data underlying figure panels in this paper may have individual authorship assigned. Where available, figure panel/source data authorship is listed in the following database record: biostudies:S-SCDT-10_1038-S44321-024-00101-6.

## Funding

## Disclosure and competing interests statement

The authors declare no competing interests.

# Expanded View Figure

**Figure EV1.　Prepatency, parasitemia curves and survival of C57BL/6 mice infected with 1000 intraerythrocytic parasites or 10,000 sporozoites of PbA or *lap(-)*.**　▶

(A) Percentage of blood-stage negative C57BL/6 mice infected with either 1000 PbA or *lap(-)* iRBC ($n = 4$ each) or 10,000 PbA or *lap(-)* sporozoites ($n = 4$ each). All C57BL/6 became infected between d4 and d7. (B) Parasitemia of mice ($n = 4$) infected in (A). Parasitemia is shown as mean parasitemia. Left panel: 2 of 4 mice infected with 1000 PbA iRBC died of ECM and 2 were not developing ECM and were dying on day 18 post-infection. All mice infected with 1000 *lap(-)* iRBC cleared the infection. Right panel: Mean parasitemia of all mice infected with 10,000 PbA or *lap(-)*sporozoites. Mean parasitemia values ± standard deviations from all infected mice are shown. (C) Rapid murine coma and behavioral score (RMCBS) of all mice ($n = 4$) infected in (A). Two of four mice infected with 1000 PbA iRBC (left panel) or 10,000 PbA sporozoites (right panel) were developing ECM symptoms starting on day 6 post-infection. All other mice were not showing a decrease of RMCBS. Mean RMCBS values ± standard deviations from all infected mice are shown. (D) Body weight of all mice ($n = 4$) infected in (A). Mice infected with 1000 PbA iRBC (left panel) or 10,000 PbA sporozoites (right panel) and developing ECM also displayed a loss of body weight until death. Two mice infected with 1000 PbA iRBC (left panel) but not developing ECM initially lost weight but then recovered followed by another loss in weight until dying on day 18 post-infection. In contrast all mice infected with either 1000 *lap(-)* iRBC (left panel) or 10,000 *lap(−)* sporozoites (right panel) recovered from temporary body weight loss. Mean body weight ± standard deviations from all infected mice is shown. (E) Survival of mice ($n = 4$) infected in (A). Mice infected with 1000 PbA iRBC died either on day 9 post-infection due to ECM or on day 18 (left panel) due to anemia. Mice infected with 10,000 PbA sporozoites (right panel) died between day 8 and 9 post-infection due to ECM. All mice infected with either 1000 *lap(−)* iRBC (left panel) or 10,000 *lap(-)* sporozoites (right panel) cleared the infection and survived. (F) Summary of PbA wild-type and *lap(−)* infections of C57BL/6 mice ($n = 4$) infected with 1000 iRBCs or 10,000 sporozoites. Numbers for peak parasitemia and days are averages from the indicated number of investigated mice. Source data are available online for this figure

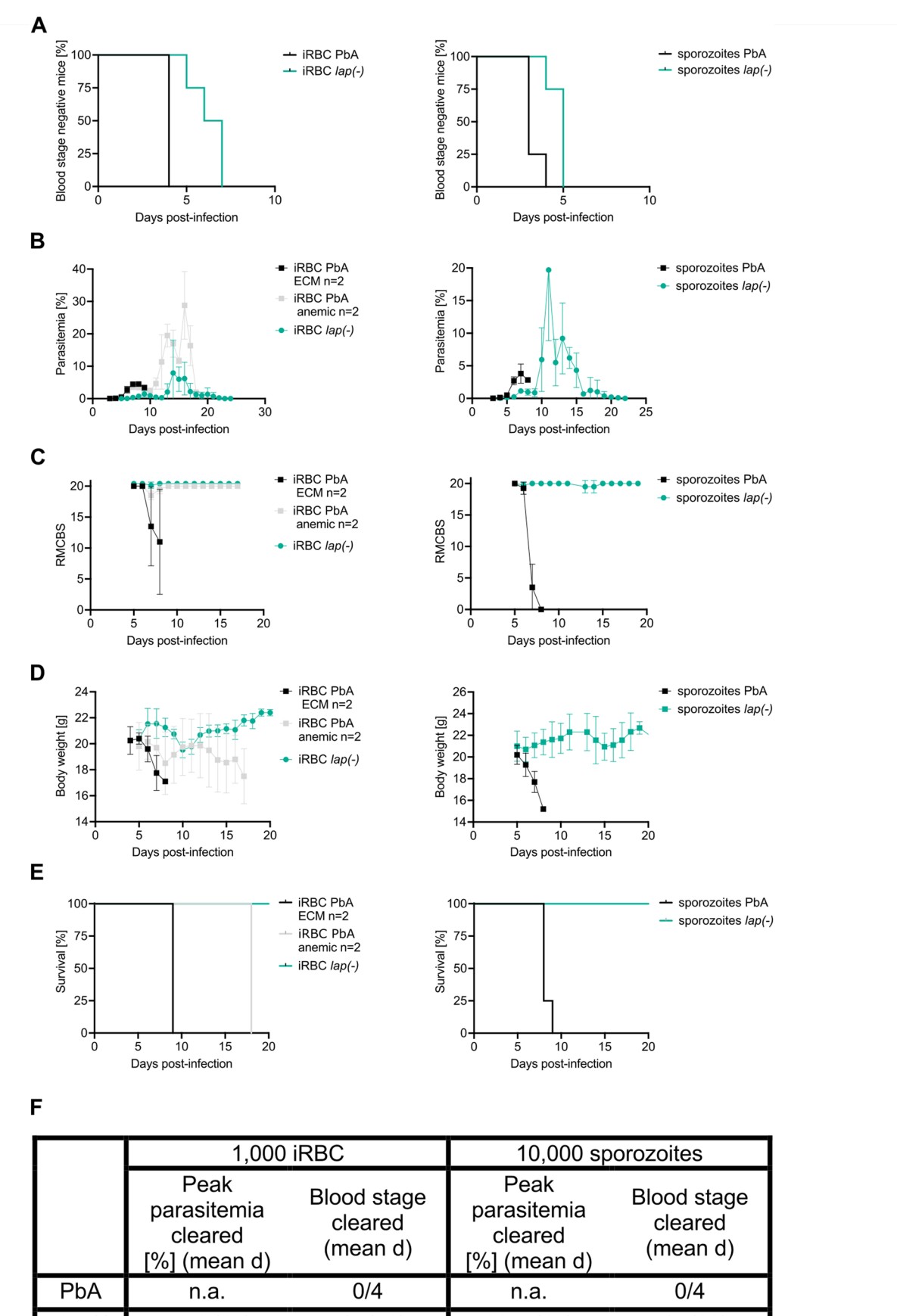

