## [Peer Review File · EMBO Molecular Medicine]

Experimental vaccination by single dose sporozoite injection of blood-stage attenuated malaria parasites

Julia Sattler, Lukas Keiber, Aiman Abdelrahim, Xinyu Zheng, Martin Jäcklin, Luisa Zechel, Catherine Moreau, Smilla Steinbrück, Manuel Fischer, Chris Janse, Angelika Hoffmann, Franziska Hentzschel, and Friedrich Frischknecht

Corresponding author: Friedrich Frischknecht (freddy.frischknecht@med.uni-heidelberg.de)

Review Timeline:

Submission Date:	27th Apr 23
Editorial Decision:	5th Jun 23
Revision Received:	22nd Apr 24
Editorial Decision:	6th Jun 24
Revision Received:	28th Jun 24
Accepted:	2nd Jul 24

Editor: Zeljko Durdevic

Transaction Report:

5th Jun 2023

Dear Prof. Frischknecht,

Thank you for the submission of your manuscript to EMBO Molecular Medicine. We have now received feedback from the three reviewers who agreed to evaluate your manuscript. All three referees recognize potential interest of the study but also raise important and partially overlapping criticism that should be addressed in a major revision. If you would like to discuss further the points raised by the referees, I am available to do so via email or video. Let me know if you are interested in this option.

We would welcome the submission of a revised version within three months for further consideration. Please let us know if you require longer to complete the revision.

I look forward to receiving your revised manuscript.

Yours sincerely,

Zeljko Durdevic

We require:

- 1) A .docx formatted version of the manuscript text (including legends for main figures, EV figures and tables). Please make sure that the changes are highlighted to be clearly visible.
- 2) Individual production quality figure files as .eps, .tif, .jpg (one file per figure). For guidance, download the 'Figure Guide PDF': (<https://www.embopress.org/page/journal/17574684/authorguide#figureformat>).
- 3) A .docx formatted letter INCLUDING the reviewers' reports and your detailed point-by-point responses to their comments. As part of the EMBO Press transparent editorial process, the point-by-point response is part of the Review Process File (RPF), which will be published alongside your paper.
- 4) A complete author checklist, which you can download from our author guidelines (<https://www.embopress.org/page/journal/17574684/authorguide#submissionofrevisions>). Please insert information in the checklist that is also reflected in the manuscript. The completed author checklist will also be part of the RPF.
- 5) Please note that all corresponding authors are required to supply an ORCID ID for their name upon submission of a revised manuscript.
- 6) It is mandatory to include a 'Data Availability' section after the Materials and Methods. Before submitting your revision, primary datasets produced in this study need to be deposited in an appropriate public database, and the accession numbers and

database listed under 'Data Availability'. Please remember to provide a reviewer password if the datasets are not yet public (see <https://www.embopress.org/page/journal/17574684/authorguide#dataavailability>).

13) Author contributions: You will be asked to provide CRediT (Contributor Role Taxonomy) terms in the submission system. These replace a narrative author contribution section in the manuscript.

14) A Conflict of Interest statement should be provided in the main text.

15) Every published paper now includes a 'Synopsis' to further enhance discoverability. Synopses are displayed on the journal

webpage and are freely accessible to all readers. They include a short stand first (maximum of 300 characters, including space) as well as 2-5 one-sentences bullet points that summarizes the paper. Please write the bullet points to summarize the key NEW findings. They should be designed to be complementary to the abstract - i.e. not repeat the same text. We encourage inclusion of key acronyms and quantitative information (maximum of 30 words / bullet point). Please use the passive voice. Please attach these in a separate file or send them by email, we will incorporate them accordingly.

Please note: When submitting your revision you will be prompted to enter your funding and payment information. This will allow Wiley to send you a quote for the article processing charge (APC) in case of acceptance. This quote takes into account any reduction or fee waivers that you may be eligible for. Authors do not need to pay any fees before their manuscript is accepted and transferred to the publisher.

EMBO Press participates in many Publish and Read agreements that allow authors to publish Open Access with reduced/no publication charges. Check your eligibility: <https://authorservices.wiley.com/author-resources/Journal-Authors/open-access/affiliation-policies-payments/index.html>

***** Reviewer's comments *****

Referee #1 (Comments on Novelty/Model System for Author):

The model is adequate and criticism of the technique is provided in the comment to the authors. The medical impact is somewhat speculative since the rodent models are not necessarily predictive of the in vivo situation in humans.

Referee #1 (Remarks for Author):

The authors present an impressive amount of work in which they screened a subset of *P. berghei* ANKA lines in which gene deletion had led to a reduced growth rate in the blood of infected mice. The aim was to obtain virulence-attenuated lines that are additionally capable of completing their life cycle through the mosquito vector, which could ultimately act as live vaccines. This was successfully achieved by the identification and isolation of two *P. berghei* ANKA clones from this subset.

The authors then clearly demonstrated reduced virulence, with no mortality and self-resolution of the patent infection and used these lines to vaccinate naïve mice which were then protected from parasite challenge (both erythrocytic and by mosquito bite). Finally, the authors applied a similar approach to CRISPR-Cas9 mediated gene-deleted *P. falciparum* parasites to obtain clones that had a reduced growth rate in vitro, though the ability of these lines to transmit was not assessed (investigation of gametocyte production was mentioned in the Materials and Methods, but I must have missed the result).

Although the data for the berghei lines is clear, I have some concerns listed below with the interpretation of the data.

- Blood stage and Sporozoite infection (Fig 2): I think it would be more correct to state that the parasitaemia of the mutant parasites was delayed, rather than saying that their growth rate was reduced, because the parasitaemia did reach high levels (Fig, 2B, 2E, 2H), moreover within a short time from low levels to peak parasitaemia. This is also the case in Fig S8. Thus, the clearance can then be partly attributed to the acquired immunity. The authors might find it better to represent parasitaemia on a log scale, a biologically more accurate representation of the course of parasitaemia.

- Protection from a homologous challenge (Fig 3): It is demonstrated that a prior infection will lead to partial protection from a homologous challenge. Thus, it is unsurprising that mice that had survived the initial infection with the mutant lines fully or partially controlled the homologous challenge. However, the authors did not demonstrate that the mice that had undergone a primary infection with the mutant line had truly cleared the infection and that they were free of parasites on the day(s) of the homologous challenge. A mere examination of a Giemsa-stained blood smear is totally inadequate for assessing true negativity (if 10 000 RBC are examined, ca. 50 fields at x 1000, a usual cut-off when estimating parasitaemia, then the detection of 1 parasite would be equivalent to a parasitaemia of ca. 500 P/μL or 0.01% assuming 5 million RBC/μL of blood). The presence or absence of parasites should have been assayed for by a sensitive PCR amplification in aliquots of at least 10 μL of blood, or by sub-inoculation. Indeed, it was previously demonstrated that *P. berghei* infections can be maintained at microscopically undetectable levels for periods exceeding 300 days, and that protection from challenge in such mice is significantly higher than in mice where parasites cannot be demonstrated. In this context, the observations of Eling and Jerusalem (see Eling 1980 Exp. Parasitol. 49 p89 and its references) should be quoted and discussed. The use of the term "sterile protection" should be

validated with a demonstration that the protected mice do not harbour sub-microscopic infections following the challenge. This in itself does not invalidate the conclusion that virulence-attenuated parasites are potentially effective vaccines, but it must be discussed especially if this is to be translated to human vaccination.

- Protection from a heterologous challenge (Fig 3): The cross-species protection (against *P. yoelii*) is of interest and a relatively rare example for malaria parasites. A somewhat similar observation that was made with respect to protection from experimental cerebral malaria due to *P. berghei* (Voza et al. 2005 Infect. Immun. 73:4777) is worth discussing, especially since this was restricted to this species but not *P. vinckei*.

Referee #2 (Comments on Novelty/Model System for Author):

The experiments are well performed.
A protective malaria blood stage vaccine is urgently needed.
Attenuated blood stages have been described previously.
The authors study both rodent and human malaria parasites.

Referee #2 (Remarks for Author):

In this manuscript by Sattler et al., the authors use a rodent malaria model to demonstrate that a single immunization with an attenuated blood stage parasite can engender protection against a pre-erythrocytic stage challenge. The authors screened 50 gene-candidates in *P. berghei* to identify genes that result in a blood stage growth defect. They obtained clonal lines from 26/50 of these gene candidates, out of which 17 genes displayed a blood stage growth defect. 2/17 gene candidates (*Pb app-* and *lap-*) were able to infect mosquitoes and generate salivary gland sporozoites, albeit at lower rates compared to WT. Remarkably, a single immunization of mice with *Pb* knockout sporozoites by IV route or by mosquito bite or blood stage challenge, was able to protect against a lethal *Pb* and a lethal heterologous *Py* challenge 6 months after immunization in both BL/6 and SW mice. However, knockout of the orthologous genes for *Pb app* and *lap* could not be generated in *P. falciparum* as previously observed from other studies, due to essentiality of these genes in blood stages. 2/6 gene deletion mutants in *P. falciparum* indicated growth defects and a potential to pursue these as whole blood stage vaccine candidates. The paper makes an important contribution to the field and provides novel insights. However, a number of additional experiments are needed to fully support the conclusions and enhance the broad impact of the paper for publication.

Major Comments

1. It is known that infection of mice with non-lethal *P. yoelii* blood stage results in protection against future sporozoite or blood stage challenge. This is likely mediated by antibodies against blood stage antigens and against antigens shared by both blood and liver stages and additionally by T cell responses in a cytokine-dependent manner (Belhoue et al. 2008, Tumwine-Downey et al 2023). However, this phenomenon is unique for non-lethal rodent malaria models because we know that in humans living in region of high malaria endemicity carrying blood stage parasites are not protected from re-infection. Gene knockouts of *app* and *lap* in lethal *P. berghei* ANKA is basically resulting in a non-lethal version of *P. berghei* (similar to *P. berghei* NK65) similar to other non-lethal rodent malaria parasites, which can protect against blood stage or sporozoite infection. Therefore, this phenomenon can be restricted to the rodent model only. To investigate this further, the authors should immunize with *P. berghei* NK65 attenuated blood stages and challenge with lethal *P. berghei* ANKA sporozoite and blood stages and evaluate protection.

2. The authors mention that the *app(-)* and *lap(-)* parasites do not have a reduced number of merozoites/schizont (Fig 1J). Is this in the first replication cycle after being released from the liver?

It is also mentioned that the *lap(-)* parasites were morphologically severely altered and appear immature by Giemsa staining (Fig 1K, no staining micrograph included to show the phenotype). These two statements are contradictory. Does the *lap(-)* generate normal number of schizonts after exo-erythrocytic stage merozoites are released from the liver and then in subsequent asexual replication cycles they start developing a growth defect?

3. There is no mechanism provided for protection following immunization with the *P. berghei* mutants to know how parasites are getting cleared.

- The authors should analyze antibody-based protection by various approaches; by ELISAs against blood stage antigens, antigens shared between liver and blood stages, reactivity of IgG isolated from immunized mice against blood stages in immunofluorescence assays, by using B cell-deficient mice etc.
- The authors should also examine the role of T cell responses in protection by blocking CD4 and CD8 T cells before challenge.
- It would be important to know the level of liver burden by qPCR in the immunized mice after sporozoite challenge to know if antibodies neutralize sporozoite entry into the liver.
- In immunized mice that are blood stage negative following sporozoite challenge, is there release of exo-erythrocytic stage merozoites? This should be verified by qPCR.
- Levels of inflammatory cytokines such as IFN γ following immunization.

4. There is incomplete analysis of *P. falciparum* mutant lines. If the authors are interested in pursuing these candidates as vaccines administered as sporozoites, they should carry detailed analysis of their ability to generate gametocytes, infect mosquitoes and generate optimum numbers of salivary gland sporozoites to make this approach feasible.

5. In the ECM model with the P. berghei mutants, the authors challenged mice with 100 blood stage parasites or bites of 10 infected mosquitoes. Have the authors done a dose-dependent study to see if 5-to-10-fold higher numbers of parasites result in ECM?

Minor comments

1. "We mostly used the gene-deletion DNA constructs..", please be clear for how many you did and for how many you did not.
2. "Around half of these 17 clonal mutant parasite lines showed reduced blood stage". Please state exact number.
3. Does set 1 and set 2 genes corresponds to Fig 1C? So 17/50 (as set 1 and 2) are represented in Fig 1C? This is not clear from the text.
4. Fig S3: What do the dots represent? Individual mice? Some knockouts have more replicates than others. Please clearly mention how many mice were used for each knockout. Overall, for all the figure legends, it should be clearly states how many mice were used for each timepoint and what statistical analysis was used.
5. Fig S5 - Some of the figures (E, G and J) are missing error bars. Please state clearly how many mice were used for the analysis
6. Also indicate salivary gland sporozoites/mosquito for the Pb app(-) and lap (-)? This is very relevant for the mosquito bite immunizations.
7. In tables describing immunizations, please mention immunizing dose.
8. "Importantly, the CRISPR-Cas9-based approach used in this study results in generation and selection of gene-deletion mutant lines that lack residual foreign genetic material used to delete the target genes, for example drug-selectable markers, or sequences of the CRISPR/Cas9 constructs." Please explain the approach clearly in the methods or in the supplementary figure. Fig S9a does not mention Pf CRISPR/Cas9 targeting strategy. Please correct this. The approach shown in Fig S10A and B so have drug selectable marker and recombination arms.
9. Entire Fig 4: Figs 4D and E do not exist but are referenced in the text.

Referee #3 (Remarks for Author):

In the manuscript titled "Vaccination by single dose sporozoite injection of blood stage attenuated malaria parasites" Sattler et al. search for Plasmodium mutants with a growth deficiency in the blood-stage of infection to use as a vaccine strategy. Although the idea and work is conceptually valid, I believe the manuscript has currently important shortcoming that is important to address.

In the introduction, the authors provide the reader with a review of genetically attenuated immunisation attempts, but lack to describe what is known regarding immunity building in populations naturally exposed to malaria, and most importantly what branch of immunity are the authors trying to induce with their approach.

Accordingly I would recommend showing in one of the mouse models used throughout the manuscript the mechanism that is promoting low parasitaemias and clearance in primary infections with the mutant parasites, and what is rendering protection after challenge with WT parasites following immunisation with the mutants.

In malaria endemic areas, multiple Plasmodium infection and clinical malaria events are known to be necessary to achieve naturally acquired immunity to malaria. Do the authors have any indication that their strategy would be more than a substitution of one malaria case? And what impact that would the immunisation planed have on children who already presented with one or a few malaria episodes?

Protection after 180d challenge was better in the app(-), while after 360d it was better for lap(-). I recommend discussing.

It is unclear to me if the cross-immunity observed in Table 2 is driven by the slow-growing blood stage of the immunising mutants or by the liver-stage. The authors should clarify, and perform immunisation with blood-stage only to disentangle the two effects.

I find a very strong disconnection between data with the mouse and human parasites. There seems to be a plot missing in figure 4 that would be the only experiment done in parallel with a mutation tested in both mouse and human Plasmodium parasites.

Minor comments:

I recommend including line number to help guidance by the reviewers.

In the intro "develop into red blood cell (RBC) invading merozoites" should be rephrased or at least add an hyphen.

"These so-called subunit vaccines are composed of..." it is unclear what so-called means here, I suggest deleting the apparently unnecessary word.

"Knobs are required for correct display of the polymorphic adhesion ligand P. falciparum erythrocyte membrane protein 1 (PfEMP1), a key virulence determinant." needs a REF

Fig S3 the authors should clarify what each circle means, Is it a mouse? figure legend should state the number of replicates in each condition. Also, the authors should clarify why some mutants appear not to have replicates.

How did you note a prolonged intraerythrocytic developmental time in vitro compared to wild type of lap(-) parasites?

How did you calculate Fig. 1J if you see no schizonts?

Lap (-) is not visible in Fig S7, I recommend using dashed lines or nudge them to see multiple overlapping lines.

The numbers 38 and 36 mice referred to in section "Sporozoite induced blood stage infections with app(-) and lap(-) are resolved by mice" should be clarified. I believe it includes all C75 and Swiss together for each condition. State it clearly in the text or include those number in the table to help the reader.

The number of Swiss mice used in Fig S8 should be stated in the fig legend.

How were signs of disease determined? Was weight of mice infected with mutants determined over time. I would be cautious to state that no signs of disease were seen, and mention only the ECM signs if that was the only measured.

The number of mice imaged in Fig 2J should be stated in fig legend.

Fig 4 includes only A-C plots but authors mention Fig4 D and E also, please revise.

The last sentence in the discussion seems to have been included by mistake. ("In general, this should include a brief (1-2 paragraph) introduction, followed by a statement of the specific scope of the study, followed by results and then interpretations.")

The mentioned line numbers correspond to the version of the manuscript with highlighted changes.

Referee #1 (Remarks for Author):

The authors present an impressive amount of work in which they screened a subset of *P. berghei* ANKA lines in which gene deletion had led to a reduced growth rate in the blood of infected mice. The aim was to obtain virulence-attenuated lines that are additionally capable of completing their life cycle through the mosquito vector, which could ultimately act as live vaccines. This was successfully achieved by the identification and isolation of two *P. berghei* ANKA clones from this subset.

The authors then clearly demonstrated reduced virulence, with no mortality and self-resolution of the patent infection and used these lines to vaccinate naïve mice which were then protected from parasite challenge (both erythrocytic and by mosquito bite). Finally, the authors applied a similar approach to CRISPR-Cas9 mediated gene-deleted *P. falciparum* parasites to obtain clones that had a reduced growth rate in vitro, though the ability of these lines to transmit was not assessed (investigation of gametocyte production was mentioned in the Materials and Methods, but I must have missed the result).

A: We thank the reviewer for his/her appreciation of our work and the constructive comments, which we address below.

Although the data for the berghei lines is clear, I have some concerns listed below with the interpretation of the data.

- Blood stage and Sporozoite infection (Fig 2): I think it would be more correct to state that the parasitaemia of the mutant parasites was delayed, rather than saying that their growth rate was reduced, because the parasitaemia did reach high levels (Fig, 2B, 2E, 2H), moreover within a short time from low levels to peak parasitaemia. This is also the case in Fig S8. Thus, the clearance can then be partly attributed to the acquired immunity. The authors might find it better to represent parasitaemia on a log scale, a biologically more accurate representation of the course of parasitaemia.

A: We partly agree with the reviewer in the reasoning that parasitaemia is delayed. We interpret this delay as a reduction in growth rate because injection of 100 mutant iRBC leads to a delay in achieving the same level of parasitemia as wild type. We addressed this by adding the information to the sentence reading "Infection of C57BL/6 mice by bite of infected mosquitoes or by intravenous injection of 1,000 *app(-)* or *lap(-)* sporozoites or infection of SWISS mice by intravenous injection of 10,000 *app(-)* or *lap(-)* sporozoites resulted in blood stage infections but with a significant delay in blood stage patency ($p < 0.0001$, Dunnett's multiple comparisons test) and a reduced blood stage growth rate compared to mice infected with wild type sporozoites (Table 1, Fig. 2D-I, Appendix Fig. S9). Sporozoite-induced wild type infections resulted in death of all 16 mice (Fig. 2F,L, Table 1, Appendix Fig. S9)." at line 242-252. We also agree in principle that a log scale representation might be more appropriate. However, on a log scale we cannot plot "0", which is essential for our representation, hence we decided for the current linear scale and would like to keep it as is.

- Protection from a homologous challenge (Fig 3): It is demonstrated that a prior infection will lead to partial protection from a homologous challenge. Thus, it is unsurprising that mice that had survived the initial infection with the mutant lines fully or partially controlled the homologous challenge. However, the authors did not demonstrate that the mice that had undergone a primary infection with the mutant line had truly cleared the infection and that they were free of parasites on the day(s) of the homologous challenge. A mere examination of a Giemsa-stained blood smear is totally inadequate for assessing true negativity (if 10 000 RBC are examined, ca. 50 fields at x 1000, a usual cut-off when estimating parasitaemia, then the detection of 1 parasite would be equivalent to a parasitaemia of ca. 500 P/ μ L or 0.01% assuming 5 million RBC/ μ L of blood). The presence or absence of parasites should have been assayed for by a sensitive PCR amplification in aliquots of at least 10 μ L of blood, or by sub-inoculation. Indeed, it was previously demonstrated that *P. berghei* infections can be maintained at microscopically undetectable levels for periods exceeding 300 days, and that protection from challenge in such mice is significantly higher than in mice where parasites cannot be demonstrated. In this context, the observations of Eling and Jerusalem (see Eling 1980 Exp. Parasitol. 49 p89 and its references) should be quoted and discussed. The use of the term "sterile protection" should be validated with a demonstration that the protected mice do not harbour sub-microscopic infections following the challenge. This in itself does not invalidate the conclusion that virulence-attenuated parasites are potentially effective vaccines, but it must be discussed especially if this is to be translated to human vaccination.

A: Thanks for suggestion, we performed the experiment and added the results as Appendix Fig. S8, and to the text (lines 231-234). This showed that the mice indeed cleared the parasites during the observation period to a level that we cannot detect them anymore by qPCR either. This of course does not rule out a potential hidden reservoir e.g. in the spleen or bone marrow that cannot be detected by peripheral blood analysis. However, we believe that does not change the overall proof-of-concept, which is the aim of our study.

We also added a paragraph on persistence of parasites to the discussion: "Persistent infections of *P. berghei* in mice were shown to be necessary for their sustained immunity against challenge infections (Eling, 1980). While we cannot rule out subpatent infections in all our mice, we examined a subset by qPCR, proving blood stage negativity. This of course does not exclude a hidden reservoir e.g. in the spleen from which new infections might recrudescence. This possibility will be important to consider for translational studies in *P. falciparum*, where subpatent infections can occur for a long time (Portugal et al, 2016, O'Flaherty et al, 2022).", line 432-438.

Appendix Fig. S8

A**B**
Appendix Fig. S8. qPCR on blood lysates of previously infected or challenged mice. (A) qPCR on mice infected with 100 *lap(-)* iRBC *i.v.*. Although mice were blood stage negative by Giemsa-stained blood smears d21 post-infection a sub-patent infection was readily detectable by qPCR in some mice tested at d40 post-infection. At d47 post-infection we were not able to detect parasites in 6/8 mice tested. From d78 post-infection onwards all mice were negative by qPCR. **(B)** qPCR data on *lap(-)* immunized mice during immunization (-d54), before (-d9) and after (d4 and d6) challenge with with 1,000 wild type sporozoites. Importantly all mice were blood stage negative by qPCR nine days (-d9) before challenge. On d4 post-challenge 1/4 immunized mice was blood stage positive. On d6 post-challenge 2/4 mice were blood stage positive detected by qPCR as well as by Giemsa-stained blood smear. Individual dots represent individual mice. Ct values of uninfected RBC from naïve mice had a mean of 33 ± 2 . Ct values of infected animals were normalized to the highest ct value (30) of naïve mice, depicted as dotted line.

- Protection from a heterologous challenge (Fig 3): The cross-species protection (against *P. yoelii*) is of interest and a relatively rare example for malaria parasites. A somewhat similar observation that was made with respect to protection from experimental cerebral malaria due to *P. berghei* (Voza et al. 2005 Infect. Immun. 73:4777) is worth discussing, especially since this was restricted to this species but not *P. vinckei*.

A: We included a statement on this and added the reference. "Cross species protection has been observed in various studies using rodent malaria with mixed results depending on parasite species, mouse strain and (co-)infections (Voza et al., 2005; Craig et al., 2012). Yet, these studies should be treated with caution in regard to translation to human malaria (Langhorne et al., 2011).", line 419-423.

Referee #2 (Comments on Novelty/Model System for Author):

The experiments are well performed.
A protective malaria blood stage vaccine is urgently needed.

Attenuated blood stages have been described previously.
The authors study both rodent and human malaria parasites.

Referee #2 (Remarks for Author):

In this manuscript by Sattler et al., the authors use a rodent malaria model to demonstrate that a single immunization with an attenuated blood stage parasite can engender protection against a pre-erythrocytic stage challenge. The authors screened 50 gene-candidates in *P. berghei* to identify genes that result in a blood stage growth defect. They obtained clonal lines from 26/50 of these gene candidates, out of which 17 genes displayed a blood stage growth defect. 2/17 gene candidates (*Pb app*- and *lap*-) were able to infect mosquitoes and generate salivary gland sporozoites, albeit at lower rates compared to WT. Remarkably, a single immunization of mice with *Pb* knockout sporozoites by IV route or by mosquito bite or blood stage challenge, was able to protect against a lethal *Pb* and a lethal heterologous *Py* challenge 6 months after immunization in both BL/6 and SW mice. However, knockout of the orthologous genes for *Pb app* and *lap* could not be generated in *P. falciparum* as previously observed from other studies, due to essentiality of these genes in blood stages. 2/6 gene deletion mutants in *P. falciparum* indicated growth defects and a potential to pursue these as whole blood stage vaccine candidates.

The paper makes an important contribution to the field and provides novel insights. However, a number of additional experiments are needed to fully support the conclusions and enhance the broad impact of the paper for publication.

A: We thank the reviewer for his/her appreciation of the complexity and novelty of our study and the thorough as well as constructive suggestions to improve our manuscript, which we answer below.

Major Comments

1. It is known that infection of mice with non-lethal *P. yoelii* blood stage results in protection against future sporozoite or blood stage challenge. This is likely mediated by antibodies against blood stage antigens and against antigens shared by both blood and liver stages and additionally by T cell responses in a cytokine-dependent manner (Belnoue et al. 2008, Tumwine-Downey et al 2023). However, this phenomenon is unique for non-lethal rodent malaria models because we know that in humans living in region of high malaria endemicity carrying blood stage parasites are not protected from re-infection. Gene knockouts of *app* and *lap* in lethal *P. berghei* ANKA is basically resulting in a non-lethal version of *P. berghei* (similar to *P. berghei* NK65) similar to other non-lethal rodent malaria parasites, which can protect against blood stage or sporozoite infection. Therefore, this phenomenon can be restricted to the rodent model only. To investigate this further, the authors should immunize with *P. berghei* NK65 attenuated blood stages and challenge with lethal *P. berghei* ANKA sporozoite and blood stages and evaluate protection.

A: We agree that the rodent model cannot cover all aspects of human malaria and, as many researchers before, we use it here as a proof-of-concept to evaluate whether it is possible to generate blood stage attenuated parasites that can be transmitted by mosquitoes. For this reason, we were not concerning ourselves with subtle differences in the mouse species and strains and opted for the most sensitive/stringent pairing of parasite strain (*PbANKA*) with host animal (C57BL/6

mice). PbANKA and PbNK65 are both lethal with C57BL/6 mice dying mostly from ECM when infected with PbANKA and mostly from anemia if infected by PbNK65. Our genetically modified parasites, however neither cause ECM nor anemia, hence are strongly virulence attenuated. Still, the question of whether virulence attenuated NK65 could protect from ANKA challenge is a valid one. We addressed this with a new experiment by infecting C57BL/6 mice with PbNK65 blood stages and treating them with chloroquine after establishment of a solid infection. Upon this cure we challenged these mice with PbANKA and found a similar clearance of this challenge infection. However, the efficacy of protection was weaker. Interestingly, all mice became blood stage patent but also all of them cleared the infection. We included the data for infection and protection in Figure 3 as the new panels I-N.

2. The authors mention that the *app(-)* and *lap(-)* parasites do not have a reduced number of merozoites/schizont (Fig 1J). Is this in the first replication cycle after being released from the liver?

A: The data are from *in vitro* culture experiments with mixed blood stages. The parasites are taken from a mouse that was highly infected and cultivated over night to reach the mature schizont stage followed by Giemsa staining and analysis/counting of merozoites. We added a sentence to make this clearer as outlined in the answer to the next comment.

It is also mentioned that the *lap(-)* parasites were morphologically severely altered and appear immature by Giemsa staining (Fig 1K, no staining micrograph included to show the phenotype). These two statements are contradictory. Does the *lap(-)* generate normal number of schizonts after exo-erythrocytic stage merozoites are released from the liver and then in subsequent asexual replication cycles they start developing a growth defect?

A: The apparent contradiction is due to a misunderstanding of the way we analyze these experiments. Essentially, we first (Fig. 1 panel J) analyze the possible maximal number of merozoites per mature schizont. Then (panel K) we analyze how many schizonts mature. This means that the mutant matures slower intracellularly but can mature fully. This distinction should have been made clearer in the text. To make sure that it is better understood, we have added some experimental explanations: "To assess if the reduced blood stage growth of *app(-)* and *lap(-)* parasites (Lin *et al*, 2015) was due to a reduced number of merozoites produced per intraerythrocytic cycle, we cultivated iRBCs overnight *in vitro*, which allows parasites to develop into mature schizonts. Quantifying the number of merozoites per schizont showed no difference between wild type and the mutants (Fig 1J). However, this result does not rule out a potential slower intraerythrocytic development. To assess this, we injected mature schizonts into naïve mice to synchronize infection and harvested blood again once ring stage parasites developed to assay the maturation of the ensuing blood stage *in vitro*.", lines 191-199.

3. There is no mechanism provided for protection following immunization with the *P. berghei* mutants to know how parasites are getting cleared.

a. The authors should analyze antibody-based protection by various approaches; by ELISAs against blood stage antigens, antigens shared between liver and blood stages, reactivity of IgG isolated from immunized mice against blood stages in immunofluorescence assays, by using B cell-deficient mice etc.

A: Because of the fundamental differences in the immune response to parasites by rodents and humans and the design of our study as a proof-of-concept for the generation of genetically attenuated parasites, we initially argued that these important questions need to be addressed in controlled human studies. However, we appreciate that some level of insight into the immune response are warranted here. We therefore now show that B-cells are important for clearance of parasites by using C57BL/6 mice depleted of B-cells by monoclonal antibodies. These data are now included in the new Figure 4. We furthermore assess the capacity of antibodies induced by the mutants to detect antigens of infected red blood cells. These data are shown in the new Appendix Figure S13.

b. The authors should also examine the role of T cell responses in protection by blocking CD4 and CD8 T cells before challenge.

A: We addressed this question by immunizing C57BL/6 mice with *lap(-)* mutants and then depleting these immunized mice from either CD4⁺ or CD8⁺ T cells prior to challenging them with wild type PbANKA. This showed that CD4⁺ but not CD8⁺ T cells were important for protection. These data are now included in the new Figure 4.

c. It would be important to know the level of liver burden by qPCR in the immunized mice after sporozoite challenge to know if antibodies neutralize sporozoite entry into the liver.

A: As we immunize with only 1,000 spz an effect of pre-erythrocytic (spz or liver stage) immunity can be excluded with high level of confidence as we need at least 50,000 sporozoite (in single dose experiments) of liver attenuated sporozoites to observe an effect on liver burden. Therefore, we feel that it would not be ethical to conduct these experiments, i.e. we would not get the approval to do them. Importantly, even if there would be a small impact, it would not change anything in the overall concept that we present.

d. In immunized mice that are blood stage negative following sporozoite challenge, is there release of exo-erythrocytic stage merozoites? This should be verified by qPCR.

A: This is a good question, that we did ponder as well. We performed a limited number of qPCRs (Appendix Fig. S8) on mice before and after challenge with 1.000 PbA sporozoites and were not able to detect parasites in a subset of these Giemsa negative mice indicating that these mice were protected from challenge. If we would want a real quantitative statement this would necessitate the killing of large numbers of mice over several days post challenge, which we feel is not warranted. We now added a statement that we can of course not exclude that some of those remaining Giemsa negative have a sub-patent infection that could be detected by qPCR. This will clearly be important to observe in eventual human trials where obtaining the necessary quantity of blood is less problematic.

e. Levels of inflammatory cytokines such as IFN γ following immunization.

A: We agree that this would be interesting to know. It is well known from literature that protection is mainly conferred by cytotoxic T-cells (liver stages) or by antibodies (blood stages) as well as pro-inflammatory cytokines. Therefore we analyzed the

levels of cytokines after immunization with *lap(-)* parasites and found that indeed the pro-inflammatory cytokines IFN γ , TNF- α , IL-1 β , IL-12p70, IL-17A, MCP-1 and IL-23 were indeed significantly increased compared to naïve mice (Figure 5) and might therefore be involved in modulating antibody-mediated clearance of *lap(-)* infections and protection from wildtype challenge (lines 359-368).

4. There is incomplete analysis of *P. falciparum* mutant lines. If the authors are interested in pursuing these candidates as vaccines administered as sporozoites, they should carry detailed analysis of their ability to generate gametocytes, infect mosquitoes and generate optimum numbers of salivary gland sporozoites to make this approach feasible.

A: We are aware that the level of analysis we provide for *P. falciparum* is limited. We provide growth attenuation data for the common lab strain 3D7. Clearly, we will now need to establish *P. falciparum* infection in our insectary first, which is an ongoing activity. To this end we use NF54 parasites and have obtained a first set of infected mosquitos. Yet we still need to consolidate these infection experiments. Next, we will have to generate transgenic NF54 lines, which is also ongoing. Following, we will need to make sure that the mutant lines, which have to be grown *in vitro* for long periods, can still infect mosquitoes. We hope the reviewer appreciates that this is a long journey and hence beyond the scope of the current proof-of-concept manuscript.

5. In the ECM model with the *P. berghei* mutants, the authors challenged mice with 100 blood stage parasites or bites of 10 infected mosquitoes. Have the authors done a dose-dependent study to see if 5-to-10-fold higher numbers of parasites result in ECM?

A: We now added experiments using 10 times these numbers, i.e. 1,000 blood stages and 10,000 sporozoites. Importantly, the overall effect is the same with no observable signs of ECM but a slightly increased peak parasitemia. We present the new data in the new Figure EV3 and in the text "In addition we assessed infections by higher doses of 1,000 *lap(-)* iRBCs and 10,000 *lap(-)* sporozoites (Fig. EV3). These mice became blood stage positive at day 5 and day 4, respectively, showed peak parasitemias of 11% and 21% at day 15 and day 12, respectively and cleared the infection by day 23 and 20, respectively (Fig. EV3), lines 267-273.

Minor comments

1. "We mostly used the gene-deletion DNA constructs..", please be clear for how many you did and for how many you did not.

A: We added a paragraph to the manuscript clearly stating how many parasite lines were generated using the constructs from PlasmoGEM and which parasite lines were previously generated. It now reads "From this screen we selected 47 gene-deletions with a predicted reduced growth rate either by 40-60% (set 1, 11 genes) or by 60-80% (set 2, 36 genes). In addition, we selected one gene-deletion mutant, *thioredoxin 2 (trx2)*, with a predicted growth rate of 0.75. For the 48 selected genes we used the gene-deletion DNA constructs available from the PlasmoGem resource to generate and select clonal mutant parasites (Fig. 1C, Appendix fig. S1, Appendix fig. S2, Appendix Tables S1 - S4). Of the 48 targeted genes, we were able to select populations of mixed wild type and transgenic parasites for 24 genes (Appendix table S1). From those populations we could isolate transgenic clonal lines for 15 genes. Of

these genes one was *trx2*, 6 genes were from set 1 and 8 genes from set 2 (Fig. 1C, Appendix Table S2). We further tested three slow growing parasite lines lacking *plasmepsin IV* (*pmlV*), *aminopeptidase p* (*app*) and *M17 leucyl-aminopeptidase* (*lap*) (Spaccapelo *et al*, 2010; Lin *et al*, 2015) - lines 153-165.

2. "Around half of these 17 clonal mutant parasite lines showed reduced blood stage". Please state exact number.

A: We would have loved to put clear and meaningful numbers here, however this would not be appropriate as the growth rate was determined differently by PlasmoGEM than by us, by necessity of the screen. Therefore, we are left with providing only a vague statement that encourages the reader to have a close look at the Appendix Figure S3 and Appendix Table S5 that lists exact numbers and describes the different ways of determining growth rates. We also added more detail to the figure legend to make these more understandable.

3. Does set 1 and set 2 genes corresponds to Fig 1C? So 17/50 (as set 1 and 2) are represented in Fig 1C? This is not clear from the text.

A: We rephrased the manuscript to make this statement clearer to the reader and it now reads "From those populations we could isolate transgenic clonal lines for 15 genes. Of these genes one was *trx2*, 6 genes were from set 1 and 8 genes from set 2 (Fig. 1C, Appendix Table S2)." lines 161-163. We also stated the numbers of genes selected for set 1 and set 2 and it now reads "From this screen we selected 47 gene-deletions with a predicted reduced growth rate either by 40-60% (set 1, 11 genes) or by 60-80% (set 2, 36 genes). In addition, we selected one gene-deletion mutant, *thioredoxin 2* (*trx2*), with a predicted growth rate of 0.75." (lines 153-156).

4. Fig S3: What do the dots represent? Individual mice? Some knockouts have more replicates than others. Please clearly mention how many mice were used for each knockout. Overall, for all the figure legends, it should be clearly states how many mice were used for each timepoint and what statistical analysis was used.

A: We assessed growth rates during limiting dilutions where mice were infected with only one parasite per mouse. Hence, the number of data points/ dots depend on the number of clones received per knockout line. In addition, we now also include data on analysis of growth rates from experiments of mice infected with 100 iRBC (Appendix Figure S3). We included an explanation into the figure legend to make this more comprehensive to the reader and also to address the previous question. The statistical analysis used can be found in the material and method section but we now also included a sentence in the figure legends throughout the manuscript.

5. Fig S5 - Some of the figures (E, G and J) are missing error bars. Please state clearly how many mice were used for the analysis

A: The number of mice used for this analysis can be found in the figure legend of Appendix Figure S5 and was 4 mice each. Error bars are missing in the late course of infection as only one mouse survived longer leading to only one value. We added a sign into the respective graphs to indicate the death of mice.

6. Also indicate salivary gland sporozoites/mosquito for the Pb app(-) and lap (-)? This is very relevant for the mosquito bite immunizations.

A: The average numbers of salivary gland sporozoites/ mosquito can be found in Appendix Table S5 but we also added now the numbers to the legend of Figure 2.

7. In tables describing immunizations, please mention immunizing dose.

A: A summary of the different immunizations used can be found in EV2.

8. "Importantly, the CRISPR-Cas9-based approach used in this study results in generation and selection of gene-deletion mutant lines that lack residual foreign genetic material used to delete the target genes, for example drug-selectable markers, or sequences of the CRISPR/Cas9 constructs." Please explain the approach clearly in the methods or in the supplementary figure. Fig S9a does not mention Pf CRISPR/Cas9 targeting strategy. Please correct this. The approach shown in Fig S10A and B so have drug selectable marker and recombination arms.

A: We apologize for not being clear enough on the procedure. As rightly pointed out, we do use a drug-selectable marker in the transfected plasmid that encodes the Cas9, the gRNA and the repair template (which introduces the deletion). However, our strategy is designed such that upon homologous recombination, only the deletion and frameshift mutation (or in case of the controls the silent shield mutation) is introduced into the genomic DNA. While the plasmid carrying the foreign transgenic material (drug selection marker, Cas9 etc) is still present in the here presented *P. falciparum* lines, it does not integrate into the genome, and as such, can upon release of the drug selection be lost from the parasite population. We think that this approach to generate a knockout without residual transgenic material poses an advantage considering possible future applications in humans. We have now clarified this by rephrasing the sentence as follows:

"Importantly, the CRISPR-Cas9-based approach used in this study does not result in the stable integration of foreign genetic material, such as drug-selectable markers or CRISPR/Cas9 sequences, into the genome of *P. falciparum*. *P. falciparum* mutant parasites lacking any transgenes can thus be obtained by prolonged blood-stage culture, which will result in the loss of the episomal plasmid." lines 399-404.

We also added the following sentence to the figure legend of Appendix Figure S14: "Expression of Cas9 and gRNA introduces a cleavage in the gene of interest, which is repaired by (A) a repair template designed to introduce a deletion, a frameshift and a stop coding into the gene or (B) a repair template designed to only introduce a shielding mutation. Note that only the repair template itself integrates, not the plasmid containing foreign genetic material." lines 1419-1424.

9. Entire Fig 4: Figs 4D and E do not exist but are referenced in the text.

A: Thanks for spotting this mistake. We changed the manuscript to the correct panels, now Figure 6.

Referee #3 (Remarks for Author):

In the manuscript titled "Vaccination by single dose sporozoite injection of blood stage attenuated malaria parasites" Sattler et al. search for Plasmodium mutants with a growth deficiency in the blood-stage of infection to use as a vaccine strategy. Although the idea and work is conceptually valid, I believe the manuscript has currently important shortcoming that is important to address.

A: We thank the reviewer to appreciate the importance of our concept and work and the constructive critique, which is in part overlapping with that of reviewer 2. Please find our answers below.

In the introduction, the authors provide the reader with a review of genetically attenuated immunisation attempts, but lack to describe what is known regarding immunity building in populations naturally exposed to malaria, and most importantly what branch of immunity are the authors trying to induce with their approach. Accordingly I would recommend showing in one of the mouse models used throughout the manuscript the mechanism that is promoting low parasitaemias and clearance in primary infections with the mutant parasites, and what is rendering protection after challenge with WT parasites following immunisation with the mutants.

A: Thanks for this comment, which essentially mirrors comment 3a and 3b from reviewer 2. We addressed this by including experiments detailed in the new Figure 4 and in the text, lines 331-358.

In malaria endemic areas, multiple Plasmodium infection and clinical malaria events are known to be necessary to achieve naturally acquired immunity to malaria. Do the authors have any indication that their strategy would be more than a substitution of one malaria case? And what impact that would the immunisation planed have on children who already presented with one or a few malaria episodes?

A: This is an important point. We would assume that our strategy would yield the equivalent immunization effect of one untreated infection. However, it is not clear if an infection with attenuated parasites would stimulate the immune system to the same level as a full-blown infection. Hence, we consider our strategy also as a tool to answer this question. We added a few sentences in the introduction: "Such Wbs could be used to study the immune response to controlled human infections and possibly employed for repeated vaccinations to build up blood stage immunity in analogy to natural occurring immunity in highly endemic regions (Doolan et al, 2009)." Lines 130-133.

Protection after 180d challenge was better in the app(-), while after 360d it was better for lap(-). I recommend discussing.

A: We didn't delve into this too deeply due to the low numbers of animals that we could challenge at this stage. Please note that we included also an experiment with NK65 infection followed by clearance of blood stages and challenge infection. Also, here there is an interesting difference with all challenged mice being infected (Figure 3). Again, the numbers are small but indeed this difference would be interesting to investigate further. We added the statement "These subtle differences in protective efficacy warrant further studies." to the discussion, line 430-431.

It is unclear to me if the cross-immunity observed in Table 2 is driven by the slow-growing blood stage of the immunising mutants or by the liver-stage. The authors should clarify, and perform immunisation with blood-stage only to disentangle the two effects.

A: As we only immunize with 1,000 sporozoites, which do not yield much pre-erythrocytic immunity even in fully liver stage attenuated parasites, we abstained from making this experiment as it would necessitate way too many mice for likely no significant outcome and it would not affect the proof-of-concept of our paper, see also answer to reviewer 2, comment 3c.

I find a very strong disconnection between data with the mouse and human parasites. There seems to be a plot missing in figure 4 that would be the only experiment done in parallel with a mutation tested in both mouse and human Plasmodium parasites.

A: We agree in principle. But we hope the reviewer appreciates that we spend a lot of time in trying to delete *lap* and *app* in *P. falciparum*, as did others and during this time added the presented experiments of the other deletions. To make this a more connected "story" we now added the deletion of *trx2* in both *P. berghei* and *P. falciparum* (Figure 1, Appendix Fig. S1, Appendix Fig. S3, Appendix Fig. S4, Appendix Fig. S5, Appendix Table 5)

Minor comments:

I recommend including line number to help guidance by the reviewers.

A: Apologies, added now and used for referring our answers.

In the intro "develop into red blood cell (RBC) invading merozoites" should be rephrased or at least add an hyphen.

A: Changed into "develop into merozoites, which invade red blood cell (RBC)." Line 68-69

"These so-called subunit vaccines are composed of..." it is unclear what so-called means here, I suggest deleting the apparently unnecessary word.

A: Thanks, we deleted "so-called", line 75

"Knobs are required for correct display of the polymorphic adhesion ligand P. falciparum erythrocyte membrane protein 1 (PfEMP1), a key virulence determinant." needs a REF

A: We added here: Crabb et al. Cell: Targeted Gene disruption shows that knobs enable malaria-infected red cells to cytoadhere under physiological shear stress. 1997. Line119

Fig S3 the authors should clarify what each circle means, Is it a mouse? figure legend should state the number of replicates in each condition. Also, the authors should clarify why some mutants appear not to have replicates.

A: One dot indicates the growth rate of an individual mouse infected with one clonal parasite. We now expanded this figure with additional data from different infection experiments and added a more comprehensive description to the figure legend Fig S3 clarifying that.

How did you note a prolonged intraerythrocytic developmental time in vitro compared to wild type of *lap(-)* parasites?

A: We infected mice with merozoites leading to a synchronized infection. The blood of these mice was collected two hours post-infection and transferred to an in vitro culture. From this culture samples were taken every two hours starting 24h post-infection and developmental stage of the parasites was assessed via Giemsa-stained blood smear. Parasites were classified as trophozoites, early and late schizonts. In contrast to wild type not late schizonts were visible at 26h post-infection in *lap(-)* infected cultures.

How did you calculate Fig. 1J if you see no schizonts?

A: To calculate the number of merozoites per schizonts blood of highly infected mice was collected and transferred to an overnight in vitro culture. The next day schizonts were purified and the number of merozoites per schizont was analysed on Giemsa-stained blood smears. *Lap(-)* blood stages were able to mature in vitro when mixed blood stages directly from an infected mouse was used. In contrast, in in vitro cultures inoculated with blood infected with synchronized *lap(-)* ring stages (Fig 1K) we never observed mature schizonts indicating that an essential factor needed for maturation is missing in the media.

Lap (-) is not visible in Fig S7, i recommend using dashed lines or nudge them to see multiple overlapping lines.

A: All data for *lap(-)* infections in C57BL/6 can be found in Fig 1. Infections of SWISS mice infected with 100 iRBC can be found in Appendix Fig S5-S6. For better visibility we nudged Fig S6 A and B and hope that the data are now better displayed.

The numbers 38 and 36 mice referred to in section "Sporozoite induced blood stage infections with *app(-)* and *lap(-)* are resolved by mice" should be clarified. I believe it includes all C75 and SWISS together for each condition. State it clearly in the text or include those number in the table to help the reader.

A: We now rephrased the sentence and it now reads "In contrast, out of 44 *app(-)* sporozoite infected mice only one C57BL/6 mouse infected with *app(-)* sporozoites died (Fig. 2F,I, Table 1, Fig. EV2, Appendix Fig. S9). All 69 mice infected with *lap(-)* sporozoites survived (Fig. 2F,I, Table 1, Fig. EV2, Appendix Fig. S9)." Line 249-252. In addition, we generated a supplementary table summarizing all mice infected with route of infection and dying or clearing, Figure EV2.

The number of SWISS mice used in Fig S8 should be stated in the fig legend.

A: Thanks for mentioning this. We added the number of SWISS mice infected with 10,000 sporozoites into the figure legend of now Fig S9

How were signs of disease determined? Was weight of mice infected with mutants determined over time. I would be cautious to state that no signs of disease were seen, and mention only the ECM signs if that was the only measured.

A: We added a figure into the supplementary data displaying assessment of RMCBS as well as weight over the course of infections of PbA and *lap(-)* infected C57BL/6. These data can be found in Figure EV3.

The number of mice imaged in Fig 2J should be stated in fig legend.

A: We added the number of animals imaged into the figure legend of Fig 2.

Fig 4 includes only A-C plots but authors mention Fig4 D and E also, please revise.

A: We accidentally referred to an older version of the figure and corrected this now in our new manuscript, lines 374-396. (→ Fig 4 is now Fig 6)

The last sentence in the discussion seems to have been included by mistake. ("In general, this should include a brief (1-2 paragraph) introduction, followed by a statement of the specific scope of the study, followed by results and then interpretations.")

A: Uups - thanks for spotting this. We deleted this sentence from the manuscript.

6th Jun 2024

Dear Prof. Frischknecht,

Thank you for the submission of your revised manuscript to EMBO Molecular Medicine. I am pleased to inform you that we will be able to accept your manuscript pending the following final amendments:

- 1) Please address all minor comments raised by the referee #2. All concerns raised by the referee #3 should be discussed and please consider adding the mouse model in the title as suggested by this referee.
- 2) In the main manuscript file, please do the following:
 - Please address all comments suggested by our data editors listed below:
 1. Please note that a separate 'Data Information' section is required in the legends of figures 2a-i, m-o.
 2. Please note that the legend for figure EV 3f is not provided in the manuscript. This needs to be rectified.
 3. Please note that information related to n is missing in the legends of figures 2l; 6b; EV 3b-d.
 4. Although 'n' is provided, please describe the nature of entity for 'n' in the legends of figures 3g-h.
 5. Please note that the error bars are not defined in the legends of figures 3g-h, j, n; 4d-e; 6c; EV 3b-d.
 - Add up to 5 keywords.
 - In M&M, provide the antibody dilutions that were used for each antibody.
 - In M&M, statistical paragraph should reflect all information that you have filled in the Authors Checklist, especially regarding randomization, blinding, replication.
 - Indicate in legends number and nature of replicates and exact p= values, not a range, along with the statistical test used. To keep the figures "clear" some authors found providing an Appendix table Sx with all exact p-values preferable. You are welcome to do this if you want to.
 - Please merge "Funding" with "Acknowledgments".
 - Please rename "Conflict of interests" to "Disclosure Statement & Competing Interests". We updated our journal's competing interests policy in January 2022 and request authors to consider both actual and perceived competing interests. Please review the policy <https://www.embopress.org/competing-interests> and update your competing interests if necessary.
 - Author contributions: Please remove it from the manuscript and specify author contributions in our submission system. CRediT has replaced the traditional author contributions section because it offers a systematic machine-readable author contributions format that allows for more effective research assessment. You are encouraged to use the free text boxes beneath each contributing author's name to add specific details on the author's contribution. More information is available in our guide to authors:
<https://www.embopress.org/page/journal/17574684/authorguide#authorshipguidelines>
 - Rename "Data and materials availability" to "Data availability" and replace the current text with following sentence: This study includes no data deposited in external repositories.
- 3) EV Figures: Please rename EV Figures 1 and 2 to Tables EV1 and EV2, remove their legends from the manuscript file and place them in the table files. Alternatively, move the tables and their legends to the Appendix. Rename the Figure EV3 to EV1 and update the callouts of the figure and tables in the main manuscript text. Also, please remove the source data for all tables.
- 4) Funding: Please make sure that information about all sources of funding are complete in both our submission system and in the manuscript. Currently, project grant FR2140/6-1 is missing in our submission system.
- 5) Synopsis:
 - Synopsis image: Please upload the image as a high-resolution jpeg file 550 px-wide x (250-400)-px high to illustrate your article.
 - Please check your synopsis text and image before submission with your revised manuscript. Please be aware that in the proof stage minor corrections only are allowed (e.g., typos).
- 6) For more information: This space should be used to list relevant web links for further consultation by our readers. Could you identify some relevant ones and provide such information as well? Some examples are patient associations, relevant databases, OMIM/proteins/genes links, author's websites, etc...
- 7) As part of the EMBO Publications transparent editorial process initiative (see our Editorial at <http://embomolmed.embopress.org/content/2/9/329>), EMBO Molecular Medicine will publish online a Review Process File (RPF) to accompany accepted manuscripts. This file will be published in conjunction with your paper and will include the anonymous referee reports, your point-by-point response and all pertinent correspondence relating to the manuscript. Let us know whether you agree with the publication of the RPF and as here, if you want to remove or not any figures from it prior to publication. Please note that the Authors checklist will be published at the end of the RPF.
- 8) Please provide a point-by-point letter INCLUDING my comments as well as the reviewer's reports and your detailed responses (as Word file).

I look forward to reading a new revised version of your manuscript as soon as possible.

Yours sincerely,

Zeljko Durdevic

*** Instructions to submit your revised manuscript ***

- 1) a .docx formatted version of the manuscript text (including Figure legends and tables)
- 2) Separate figure files*
- 3) supplemental information as Expanded View and/or Appendix. Please carefully check the authors guidelines for formatting Expanded view and Appendix figures and tables at <https://www.embopress.org/page/journal/17574684/authorguide#expandedview>
- 4) a letter INCLUDING the reviewer's reports and your detailed responses to their comments (as Word file).
- 5) The paper explained: EMBO Molecular Medicine articles are accompanied by a summary of the articles to emphasize the major findings in the paper and their medical implications for the non-specialist reader. Please provide a draft summary of your article highlighting
 - the medical issue you are addressing,
 - the results obtained and
 - their clinical impact.This may be edited to ensure that readers understand the significance and context of the research. Please refer to any of our published articles for an example.
- 6) For more information: There is space at the end of each article to list relevant web links for further consultation by our readers. Could you identify some relevant ones and provide such information as well? Some examples are patient associations, relevant databases, OMIM/proteins/genes links, author's websites, etc...
- 7) Author contributions: the contribution of every author must be detailed in a separate section.
- 8) EMBO Molecular Medicine now requires a complete author checklist (<https://www.embopress.org/page/journal/17574684/authorguide>) to be submitted with all revised manuscripts. Please use the checklist as guideline for the sort of information we need WITHIN the manuscript. The checklist should only be filled with page numbers where the information can be found. This is particularly important for animal reporting, antibody dilutions (missing) and exact values and n that should be indicated instead of a range.
- 9) Every published paper now includes a 'Synopsis' to further enhance discoverability. Synopses are displayed on the journal webpage and are freely accessible to all readers. They include a short stand first (maximum of 300 characters, including space) as well as 2-5 one sentence bullet points that summarise the paper. Please write the bullet points to summarise the key NEW findings. They should be designed to be complementary to the abstract - i.e. not repeat the same text. We encourage inclusion of key acronyms and quantitative information (maximum of 30 words / bullet point). Please use the passive voice. Please attach

these in a separate file or send them by email, we will incorporate them accordingly.

You are also welcome to suggest a striking image or visual abstract to illustrate your article. If you do please provide a jpeg file 550 px-wide x 300-800px high.

10) A Conflict of Interest statement should be provided in the main text

11) Please note that we now mandate that all corresponding authors list an ORCID digital identifier. This takes <90 seconds to complete. We encourage all authors to supply an ORCID identifier, which will be linked to their name for unambiguous name identification.

Currently, our records indicate that the ORCID for your account is 0000-0002-8332-6668.

Link Not Available

Photos 400-800 DPI

*Additional important information regarding figures and illustrations can be found at

<https://bit.ly/EMBOPressFigurePreparationGuideline>. See also figure legend preparation guidelines:

<https://www.embopress.org/page/journal/17574684/authorguide#figureformat>

***** Reviewer's comments *****

Referee #1 (Comments on Novelty/Model System for Author):

The model is adequate and criticism of the technique is provided in the comment to the authors. The medical impact is somewhat speculative since the rodent model is not necessarily predictive of the in vivo situation in humans.

Referee #1 (Remarks for Author):

the authors have adequately addressed all the points raised in the revised manuscript.

Referee #2 (Comments on Novelty/Model System for Author):

The authors have addressed all of my comments and generated additional data to address them and substantiate their findings. The revised version of the manuscript is much improved and I recommend publication. I have a few more minor comments about issues that should be corrected.

Referee #2 (Remarks for Author):

The authors have addressed all of my comments and generated additional data to address them and substantiate their findings. The revised version of the manuscript is much improved and I recommend publication. I have a few more minor comments about issues that should be corrected.

Certain references are incorrectly cited.

a. Ln 98 - 101 - Butler et al. 2011. "Sporozoite attenuation by genetic modification rather than by radiation offers the advantage of a more homogeneous product, increased biosafety for sporozoite production and potentially increased potency"

The Butler et al. study is a comparative study between the superior efficacy of late liver stage arresting genetically attenuated parasites compared to either radiation attenuated parasites OR early arresting GAPs. It does not mention that all GAPs have increased potency to RAS. Please rephrase or use a different reference.

b. Ln 104 - Goswami et al. 2020 is not relevant here as there were no clinical trials in this study.

c. Ln 113 - the two papers mentioned here Murphy et al, 2022 and Roestenberg et al, 2020 are genetically attenuated whole sporozoite based vaccines and not whole blood stage vaccines. In Murphy et al. the immunogen was GAP3KO sporozoites and in Roestenberg et al, 2020 it was PfSPZ-GA1 sporozoites as immunogen.

Referee #3 (Remarks for Author):

The authors addressed some of the comments and request for clarity and now it is clearer that the slow parasitaemia progression of the mutants or imposed by treatment allows for humoral immunity of mice to control infection; and that the protection conferred by immunisation of mice with the lap(-) parasites relies more on CD4 than on CD8 T cells. Nevertheless in my view the authors still fail to show that slow growing, avirulent *P. falciparum* mutants may constitute a valuable tool to inform on the induction of immune responses or help developing new attenuated parasite vaccines. Despite the large amount of work shown by the authors, the manuscript has no clinical insight, nor in my view represents a new avenue of vaccine development, as the authors provide no indication that such strategy would be of any value beyond mimicking one malaria case, and suggest in their reply to the reviewers that multiple immunisations would be necessary in the clinical setting, contrary to the title of the manuscript.

That protection against wild type challenge in mice can be induced following immunization with genetically attenuated whole blood stages of rodent malaria parasites with a reduced growth rate was known and is amply cited by the authors in the introduction, so the argument of a proof of concept of this manuscript seems over emphasised to me. That the current study is able to do that with these two mutants adds limited novelty, and had been partially shown, and specially given that the 2 targeted genes are not targetable in human malaria parasites, makes the process of relying on mouse data very unreliable, as the authors also point in the reply to the reviewers. Sattler et al. are correct that much has to be considered to avoid the potential risk of causing a symptomatic blood stage infection following whole blood stage immunisation is true, but how their study helps guide that process is not clear.

I suggest being more candid in their interpretation of their data, making reference to mouse models in the title of the manuscript.

Then more specifically to the result section I have the following comments that should be made clearer on the manuscript text

The selection criteria for the inclusion of thioredoxin 2 (trx2) is unclear and should be clarified, and I recommend mention the growth rate reduction as done for others, it is misleading the mention to 0,75, when the reduction would be 25%.

On Ln 231 I suggest clearly stating how many days was the infection detection by qPCR prolonged.

Legend of Fig 4D mentions twice anti CD20 mAB=3 and I believe once it should anti CD20 mAB=1.

The description of anti CD4 and CD8 antibody experiments is a bit confusing to me, why in 4G is the % of blood stage negative mice treated with anti CD8 mAB starting at 80% on day 0 post infection (challenge)? Parasitaemias of the different groups of infected mice should be shown to the reader.

The inclusion of the trx2 ko parasites did very little for the connection between the mouse data and the *P. Falciparum* experiments, but I defer to the editor to decide whether cohesion of the story meets the standards of the publication.

***** Reviewer's comments *****

Referee #1 (Comments on Novelty/Model System for Author):

The model is adequate and criticism of the technique is provided in the comment to the authors. The medical impact is somewhat speculative since the rodent model is not necessarily predictive of the in vivo situation in humans.

Referee #1 (Remarks for Author):

the authors have adequately addressed all the points raised in the revised manuscript.

A: We thank the reviewer for his/her constructive comments and support.

Referee #2 (Comments on Novelty/Model System for Author):

The authors have addressed all of my comments and generated additional data to address them and substantiate their findings. The revised version of the manuscript is much improved and I recommend publication. I have a few more minor comments about issues that should be corrected.

A: We thank the reviewer for his/her constructive comments and support.

Referee #2 (Remarks for Author):

The authors have addressed all of my comments and generated additional data to address them and substantiate their findings. The revised version of the manuscript is much improved and I recommend publication. I have a few more minor comments about issues that should be corrected.

Certain references are incorrectly cited.

a. Ln 98 - 101 - Butler et al. 2011. "Sporozoite attenuation by genetic modification rather than by radiation offers the advantage of a more homogeneous product, increased biosafety for sporozoite production and potentially increased potency"

The Butler et al. study is a comparative study between the superior efficacy of late liver stage arresting genetically attenuated parasites compared to either radiation attenuated parasites OR early arresting GAPs. It does not mention that all GAPs have increased potency to RAS. Please rephrase or use a different reference.

A: We rephrased the sentence and it now reads "Sporozoite attenuation by genetic modification rather than by radiation offers the advantage of a more homogeneous product, increased biosafety for sporozoite production and for some genetic attenuated parasites (GAP) increased potency (Butler et al. 2011)."

b. Ln 104 - Goswami et al. 2020 is not relevant here as there were no clinical trials in this study.

A: we deleted this reference from the manuscript line 104

c. Ln 113 - the two papers mentioned here Murphy et al, 2022 and Roestenberg et al, 2020 are genetically attenuated whole sporozoite based vaccines and not whole blood stage vaccines. In Murphy et al. the immunogen was GAP3KO sporozoites and in Roestenberg et al, 2020 it was PfSPZ-GA1 sporozoites as immunogen.

A: Thanks for spotting this mistake. We deleted these references here and added Ting 2008, Spaccapelo 2010 and Aly 2010.

Referee #3 (Remarks for Author):

The authors addressed some of the comments and request for clarity and now it is clearer that the slow parasitaemia progression of the mutants or imposed by treatment allows for humoral immunity of mice to control infection; and that the protection conferred by immunisation of mice with the lap(-) parasites relies more on CD4 than on CD8 T cells.

A: Thanks for agreeing that we added substantial amounts of data to the manuscript and made our message clearer.

Nevertheless in my view the authors still fail to show that slow growing, avirulent *P. falciparum* mutants may constitute a valuable tool to inform on the induction of immune responses or help developing new attenuated parasite vaccines.

A: Here, we present a proof-of-concept study that does not aim to show immune responses to *P. falciparum*. This will be done in a follow-up study and require a few years of work. If that concept holds up in *P. falciparum*, we believe it will open up the possibility to perform prolonged (e.g. 2 to 3 times as long as with wildtype parasites) controlled human infections enabling the study of induction of the adaptive immune responses.

Despite the large amount of work shown by the authors, the manuscript has no clinical insight, nor in my view represents a new avenue of vaccine development, as the authors provide no indication that such strategy would

be of any value beyond mimicking one malaria case, and suggest in their reply to the reviewers that multiple immunisations would be necessary in the clinical setting, contrary to the title of the manuscript.

A: While we do not show any clinical data we hope that our approach could eventually contribute to clinical studies that either inform improved vaccine design or be a constitute a new vaccine approach aiming not at inducing sterile immunity but disease suppressing immunity.

That protection against wild type challenge in mice can be induced following immunization with genetically attenuated whole blood stages of rodent malaria parasites with a reduced growth rate was known and is amply cited by the authors in the introduction, so the argument of a proof of concept of this manuscript seems over emphasised to me.

A: The proof of concept concerns the immunization with sporozoites and not mixed red blood stages. This is in a number of ways, in our modest view, an improvement to the published genetically attenuated whole blood stages of rodent malaria parasites with a reduced growth rate and as our high failure rate suggests, is no easy feat to achieve. This approach might enable the possibility to perform controlled human studies with infected mosquito bites or GMP produced sporozoites thus open up the option to compare the protection efficacy directly to PfSPZ vaccine immunizations.

That the current study is able to do that with these two mutants adds limited novelty, and had been partially shown, and specially given that the 2 targeted genes are not targetable in human malaria parasites, makes the process of relying on mouse data very unreliable, as the authors also point in the reply to the reviewers.

A: Totally true and initially very disappointing to us as the reviewer might appreciate.

Sattler et al. are correct that much has to be considered to avoid the potential risk of causing a symptomatic blood stage infection following whole blood stage immunisation is true, but how their study helps guide that process is not clear.

A: The current PfSPZ e.g. LARC2 mutants could be generated in a parasite line that grows slower in the blood. In that case, there would be less worry about breakthrough infections as they would be controlled.

I suggest being more candid in their interpretation of their data, making reference to mouse models in the title of the manuscript.

A: Fair enough. We decided to add "experimental" to the title to reflect the use of the mouse model

Then more specifically to the result section I have the following comments that should be made clearer on the manuscript text

The selection criteria for the inclusion of thioredoxin 2 (trx2) is unclear and should be clarified, and I recommend mention the growth rate reduction as done for others, it is misleading the mention to 0,75, when the reduction would be 25%.

A: We rephrased the sentence, and it now reads "In addition, we selected one gene-deletion mutant, *thioredoxin 2* (trx2), with a predicted growth rate that was reduced by 25%.

On In 231 I suggest clearly stating how many days was the infection detection by qPCR prolonged.

A: We included all the qPCR data in a separate Appendix Fig. S8 as indicated. Therefore, we didn't add an additional statement into the manuscript text.

Legend of Fig 4D mentions twice anti CD20 mAB=3 and I believe once it should anti CD20 mAB=1.

A: Thanks for spotting this. We corrected the figure 4D to CD20 mAB=1.

The description of anti CD4 and CD8 antibody experiments is a bit confusing to me, why in 4G is the % of blood stage negative mice treated with anti CD8 mAB starting at 80% on day 0 post infection (challenge)?

Parasitaemias of the different groups of infected mice should be shown to the reader.

A: Thanks for spotting this. These mice definitely were blood stage negative at the start of the experiment. We checked the original data and found that we made a mistake when transferring the data of one mouse to GraphPad. We corrected now the data and generated a new figure 4G. The corrected data are showing the same trend and therefore the interpretation of the data is still valid.

We decided against showing parasitemia curves as showing % of blood stage negative mice is an easier more visual way of interpreting the data and parasitemia curves would not add more information to the data.

The inclusion of the trx2 ko parasites did very little for the connection between the mouse data and the P. Falciparum experiments, but I defer to the editor to decide whether cohesion of the story meets the standards of the publication.

A: We are aware that the connection is not as strong as we wished it could be if we would include ten knockout parasite lines but we think it is an advance to show that it is possible to generate slow-growing *Plasmodium* lines by screening *P. berghei* and in *P. falciparum*. As generation of *P. falciparum* knockout lines is elaborate and time consuming work that amplifies for slow growing ones we think that the presented data are sufficient as a proof of concept. Clearly, more *P. falciparum* slow growing parasite lines and in-depth analysis regarding mechanism of attenuation as well as transmission to mosquitoes are needed before the first controlled human infections should be envisaged.

1st Jul 2024

Dear Prof. Frischknecht,

We are pleased to inform you that your manuscript is accepted for publication and is now being sent to our publisher to be included in the next available issue of EMBO Molecular Medicine.
